# Sculpting ion channel functional expression with engineered ubiquitin ligases

Scott A Kanner[1], Travis Morgenstern[2], Henry M Colecraft[1,2,3]*

[1]Doctoral Program in Neurobiology and Behavior, Columbia University College of Physicians and Surgeons, New York, United States; [2]Department of Pharmacology, Columbia University College of Physicians and Surgeons, New York, United States; [3]Department of Physiology and Cellular Biophysics, Columbia University College of Physicians and Surgeons, New York, United States

**Abstract** The functional repertoire of surface ion channels is sustained by dynamic processes of trafficking, sorting, and degradation. Dysregulation of these processes underlies diverse ion channelopathies including cardiac arrhythmias and cystic fibrosis. Ubiquitination powerfully regulates multiple steps in the channel lifecycle, yet basic mechanistic understanding is confounded by promiscuity among E3 ligase/substrate interactions and ubiquitin code complexity. Here we targeted the catalytic domain of E3 ligase, CHIP, to YFP-tagged KCNQ1 ± KCNE1 subunits with a GFP-nanobody to selectively manipulate this channel complex in heterologous cells and adult rat cardiomyocytes. Engineered CHIP enhanced KCNQ1 ubiquitination, eliminated KCNQ1 surface-density, and abolished reconstituted $K^+$ currents without affecting protein expression. A chemo-genetic variation enabling chemical control of ubiquitination revealed KCNQ1 surface-density declined with a ~ 3.5 hr $t_{1/2}$ by impaired forward trafficking. The results illustrate utility of engineered E3 ligases to elucidate mechanisms underlying ubiquitin regulation of membrane proteins, and to achieve effective post-translational functional knockdown of ion channels.
DOI: https://doi.org/10.7554/eLife.29744.001

*For correspondence:
hc2405@cumc.columbia.edu

Competing interests: The authors declare that no competing interests exist.

## Introduction

Integral surface membrane proteins including ion channels, transporters, and receptors are vital to the survival and function of all cells. Consequently, processes that control the surface abundance and composition of membrane proteins are critical determinants of cellular biology and physiology. Impaired surface trafficking of membrane proteins underlies diverse diseases ranging from cystic fibrosis to cardiac arrhythmias (*Gelman and Kopito, 2002*; *Anderson et al., 2014*), motivating a need to better understand fundamental mechanisms controlling membrane protein surface density. The surface repertoire of membrane proteins is regulated by multi-layered maturation and trafficking processes (*MacGurn et al., 2012*; *Foot et al., 2017*). As such, the mechanisms governing diverse aspects of membrane protein fate is an intensely studied research area.

Ubiquitination determines membrane protein functional expression by regulating multiple steps in the membrane protein lifecycle. Ubiquitin is a 76-residue protein that can be covalently attached to lysine residues on polypeptide substrates through the sequential action of three enzymes: a ubiquitin activation enzyme (E1); a ubiquitin-conjugating enzyme (E2); and a ubiquitin ligase (E3), that catalyzes transfer of ubiquitin to substrates. The human genome encodes 2 E1s, 37 E2s, and >600 E3 ubiquitin ligases. Ubiquitin contains seven lysine residues (K6, K11, K27, K29, K33, K48, K63) that, together with its N-terminus (Met1), can serve as secondary attachment points to make diverse poly-ubiquitin chains with different structures and functions (*Komander, 2009*). Ubiquitination has

**eLife digest** Cells are surrounded by a membrane that separates the outside of the cell from its inside. Proteins called ion channels are embedded within this membrane and allow charged ions to move in and out of the cell. The movement of ions generates electrical currents that are essential for many processes that keep us alive, including our heartbeat and the activity within our brain.

Like many other proteins, newly made ion channels undergo several steps before they mature and become active. Cells destroy any proteins that do not mature properly, as well as those that become damaged or are simply no longer needed. A small protein called ubiquitin helps to mark such unwanted proteins for destruction. Enzymes known as E3 ligases attach ubiquitin to target proteins in a process known as ubiquitination. This process regulates both the quality and amount of proteins within cells.

To understand the role of a particular protein, it is often necessary to remove it from the cell and then examine the consequences. In the past, researchers have harnessed the ubiquitin system to remove many kinds of proteins, but this approach had not previously been used to target an ion channel.

Now, Kanner et al. set out to selectively eliminate ion channels via targeted ubiquitination. The experiments showed that previous approaches that could destroy proteins within the cell were not effective against ion channels. Kanner et al. then engineered a particular E3 ligase so that it could selectively attach ubiquitin to the desired ion channels. This approach successfully prevented the channels from reaching the cell membrane, thereby silencing the electrical currents that they normally generate. Additionally, a new tool was developed to stop ion channels in their tracks, essentially with a flip of a chemical switch. Kanner et al. then used this approach to manipulate ion channels in a highly controlled manner, within their normal environment of heart muscle cells.

These new approaches form a toolset that scientists can now exploit to study diverse ion channels. In the future, the toolkit could potentially be used to develop treatments for disorders such as epilepsy, chronic pain, and irregular heartbeats, where too many channels are active or present at the cell membrane.

DOI: https://doi.org/10.7554/eLife.29744.002

classically been ascribed to targeting cytosolic proteins for degradation by the proteasome (*Hershko and Ciechanover, 1998*). However, it is now evident that ubiquitination of both cytosolic and membrane proteins can lead to more nuanced outcomes including regulating protein trafficking/sorting, stability, and/or function (*Komander, 2009*; *Foot et al., 2017*). Nevertheless, precisely how ubiquitination regulates such diverse aspects of protein fate— and membrane protein fate in particular— is often poorly understood. Factors that complicate analyses include: (1) multiple E3 ligases may ubiquitinate a single substrate; (2) a particular E3 can typically catalyze ubiquitination of multiple substrates; (3) distinct E3 ligases can have preference for particular ubiquitination profiles (e.g. monoubiquitination versus polyubiquitination) and polyubiquitin chain linkages (e.g. K48 versus K63); (4) lack of temporal control over the ubiquitination process.

The elusive nature of ubiquitin signaling is exemplified by its regulation of diverse voltage-gated ion channels. KCNQ1 (Kv7.1; Q1), is a voltage-gated $K^+$ channel which together with auxiliary KCNE1 subunits give rise to the slowly activating delayed rectifier current $I_{Ks}$ that is important for human ventricular action potential repolarization (*Barhanin et al., 1996*; *Sanguinetti et al., 1996*). Loss-of-function mutations in Q1 lead to long QT syndrome type 1 (LQT1), a precarious condition that predisposes affected individuals to exertion-triggered cardiac arrhythmias and sudden cardiac death (*Tester et al., 2005*). In heterologous expression studies, NEDD4-2, a HECT domain E3 ligase, binds a PY motif on Q1 C-terminus; enhances Q1 ubiquitination; down-regulates Q1 expression; and inhibits $I_{Ks}$ current (*Jespersen et al., 2007*). Understanding precisely how NEDD4-2 accomplishes these distinctive effects is confounded by the promiscuity of this E3 ligase in targeting many other proteins that contain PY motifs (*Abriel and Staub, 2005*; *MacGurn et al., 2012*; *Goel et al., 2015*), as well as a lack of tight temporal control over its action. This could potentially be resolved if it were possible to target distinct E3 ligase activity to Q1/KCNE1 proteins in a selective and temporally controllable manner.

Several studies have applied an approach that utilizes engineered E3 ubiquitin ligases to selectively target cytosolic proteins to direct their degradation by the proteasome (*Zhou et al., 2000*; *Hatakeyama et al., 2005*; *Caussinus et al., 2011*; *Ma et al., 2013*; *Portnoff et al., 2014*). The general principle involves replacing the intrinsic substrate-targeting module of an E3 ligase with a motif that directs it to a desired target protein. Here, we sought to determine, for the first time, whether this method could be applied to elucidate mechanisms underlying ubiquitin regulation of ion channel complexes. We engineered E3 ubiquitin ligases to selectively target YFP-tagged Q1 or KCNE1 subunits and assessed the impact on channel surface density, stability, and $I_{Ks}$. We found that targeted ubiquitination of Q1/KCNE1 with distinct engineered ligases dramatically diminished channel surface expression without necessarily affecting total protein expression. We developed a chemo-genetic variation of the approach that enabled controllable targeting of an engineered E3 to the channel using chemical heterodimerization. The temporal control afforded by the chemo-genetic method in combination with fluorescence pulse-chase assays revealed that ubiquitination diminished Q1 surface density by selectively limiting channel delivery to the cell surface, and not by enhancing the rate of endocytosis. To demonstrate the generality of the approach, we used the engineered E3 ligase to selectively eliminate surface expression and currents through voltage-gated L-type Ca$^{2+}$ (Ca$_V$1.2) channels. Similar to Q1, targeted ubiquitination of Ca$_V$1.2 markedly decreased channel surface density without impacting total expression, emphasizing a fundamental distinction in the impact of targeted ubiquitination between ion channels and previously studied cytosolic proteins. Beyond enabling original mechanistic insights, the approach provides a potent tool to post-translationally manipulate surface expression of ion channel macromolecular complexes in a manner that complements, and provides particular advantages over, well-established and widely used genomic/mRNA interference methods.

## Results

### Design of engineered ubiquitin ligases to manipulate Q1 functional expression

The lifecycle of surface ion channels and other membrane proteins involves minimally their genesis and folding in the endoplasmic reticulum (ER); post-translational maturation in the Golgi; their delivery to and removal from their site of action on the plasma membrane; and ultimately their demise by degradation in lysosomes or via the proteasome (*Figure 1A*). Ubiquitination looms as a powerful mechanism to control membrane protein fate since it potentially influences multiple steps in their lifecycle (*Figure 1A*). Ubiquitination is mediated by a step-wise cascade of three enzymes (E1, E2, E3), resulting in the covalent attachment of the 76-residue ubiquitin to lysines of a target protein (*Figure 1B*).

We sought to develop a system that enabled selective ubiquitination of the voltage-gated K$^+$ channel pore-forming subunit, Q1, to dissect the mechanistic impact of specific post-translational modification of this protein. We took advantage of the modular design of E3 ligases, which typically have distinct substrate-binding and catalytic domains. For example, CHIP (C-terminus of the Hsp70-interacting protein), is a U-box E3 ligase comprised of a catalytic domain that binds E2 and a tetratricopeptide repeats (TPR) targeting domain that binds Hsp70 (*Connell et al., 2001*; *Murata et al., 2003*; *Zhang et al., 2005*). This modular arrangement enables its function for chaperone-mediated ubiquitination of substrate proteins as a quality control mechanism (*Figure 1B*). We substituted the TPR domain of CHIP with the vhh4 nanobody, which binds GFP/YFP (but not CFP) (*Kubala et al., 2010*), creating nanoCHIP. We hypothesized that nanoCHIP would selectively target and catalyze ubiquitination of Q1-YFP, leading to three possible (but not mutually exclusive) outcomes of reducing protein stability, altering trafficking, or modulating channel function (*Figure 1C*).

### nanoCHIP abolishes Q1 surface population, with modest effect on total channel pool

We utilized optical fluorescence assays to conveniently measure surface and total pools of Q1-YFP in a robust and high throughput manner. We introduced a 13-residue high-affinity bungarotoxin binding site (BBS) into the extracellular S1-S2 loop of Q1, enabling detection of surface channels in non-permeabilized cells with Alexa Fluor 647-conjugated bungarotoxin (BTX$_{647}$) (*Figure 1C* and *Figure 2*)

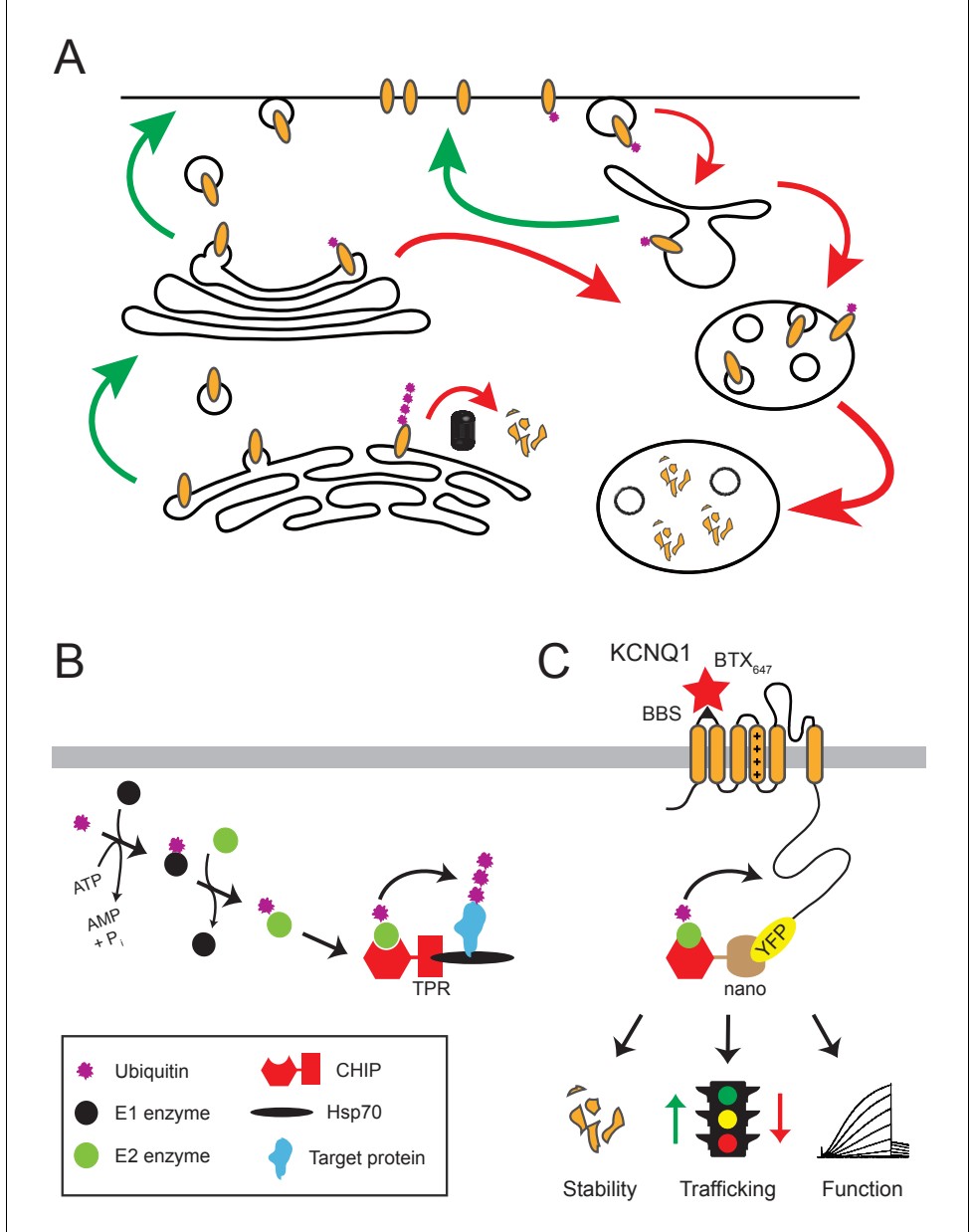

**Figure 1.** Role of ubiquitin in the lifecycle of membrane proteins. (**A**) Dynamic trafficking of membrane proteins among subcellular compartments. Degradation can take place from the ER via the proteasome, or through downstream endocytic compartments via the lysosome. Ubiquitin (purple) is a molecular signal important for mediating multiple steps in membrane protein trafficking, function, and degradation. Forward trafficking (green) and reverse trafficking (red) processes are represented. (**B**) Enzymatic cascade of ubiquitination, including the ATP-dependent activation of ubiquitin (**E1**), ubiquitin conjugation (**E2**), and ultimate ubiquitin transfer to target substrate (**E3**). CHIP is an E3 ligase, recognizing Hsp70-bound substrates via the TPR binding domain and catalyzing their ubiquitination via the U-box domain (hexagon). (**C**) Schematic for engineering an E3 ubiquitin ligase and potential outcomes on an ion channel substrate. The substrate-binding TPR domain of CHIP is replaced with GFP-binder, vhh4 nanobody, creating nanoCHIP which has novel selectivity towards YFP-tagged Q1 subunits. The bungarotoxin binding site (BBS) epitope (**S1–S2**) allows for selective labeling of surface Q1 subunits, YFP signal represents total Q1 expression. This experimental paradigm enables robust analysis of Q1 stability, trafficking, and function.

DOI: https://doi.org/10.7554/eLife.29744.003

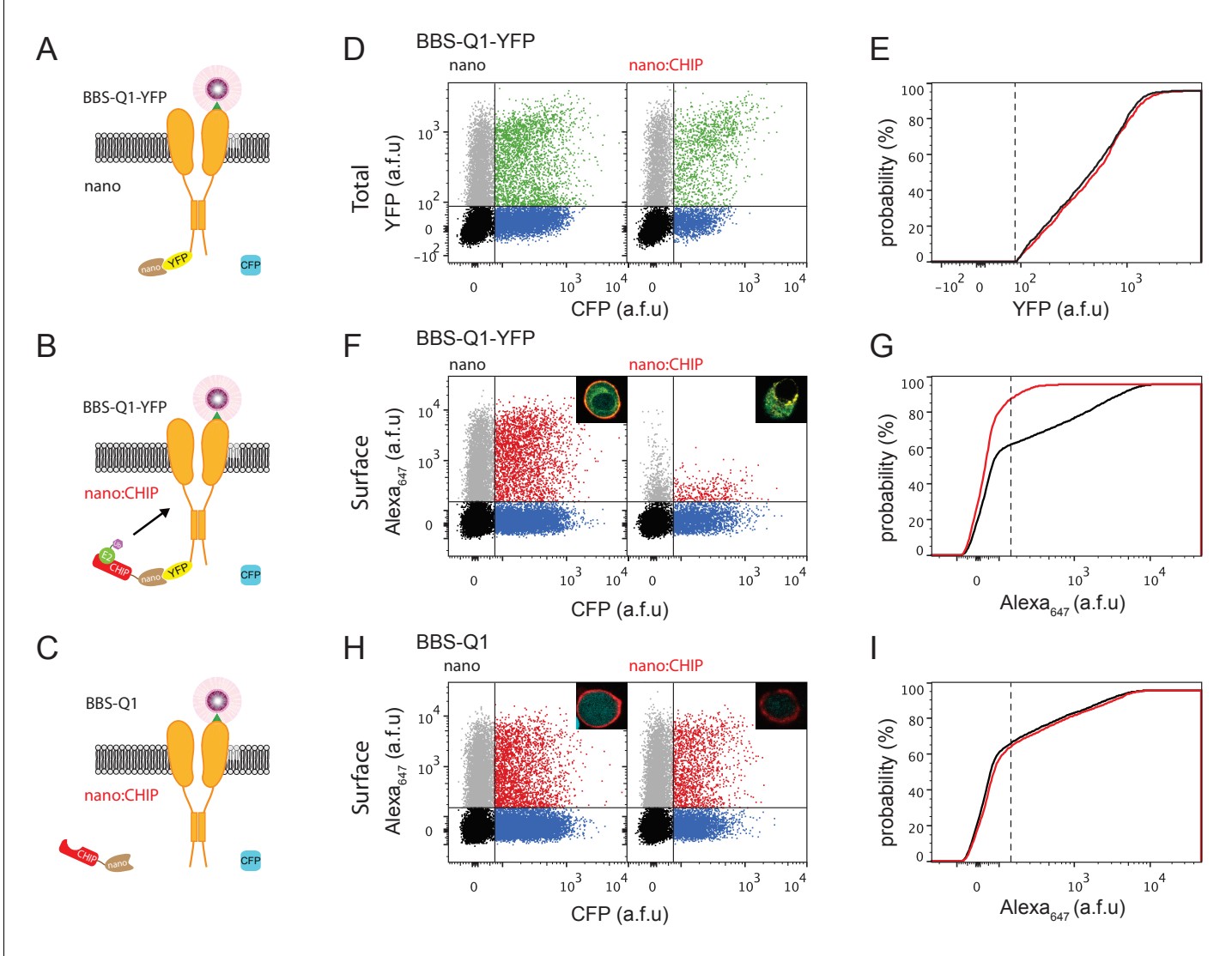

**Figure 2.** nanoCHIP selectively abolishes Q1 surface expression. (**A–C**) Cartoons of experimental strategies. BBS-Q1-YFP was co-transfected with either nanobody alone (**A**) or with nanoCHIP (**B**). Untagged BBS-Q1 co-expressed with nanoCHIP was used as a control to test for specificity of the approach (**C**). (**D**) Flow cytometry analyses of total Q1 expression (YFP fluorescence) in cells expressing BBS-Q1-YFP + nanobody (*left*, control) or BBS-Q1-YFP + nanoCHIP (*right*).~50,000 cells are represented in dot plots here and throughout. Vertical and horizontal lines represent thresholds for CFP and YFP-positive cells, respectively, based on analyses of single color controls. Represented are CFP-positive cells with YFP signal above (green dots) or below threshold (blue dots); YFP-positive cells with CFP signal below threshold (gray dots); and untransfected cells (black dots). (**E**) Cumulative distribution histograms of YFP fluorescence for BBS-Q1-YFP co-expressed with either nanobody (black line) or nanoCHIP (red line). Plot generated from population of YFP- and CFP-positive cells. Dotted line is threshold value for YFP signal. (**F**) Flow cytometry analyses of surface Q1 channels (Alexa$_{647}$ fluorescence) in cells expressing BBS-Q1-YFP + nanobody (*left*, control) or BBS-Q1-YFP + nanoCHIP (*right*). Representative confocal images are inset. (**G**) Cumulative distribution histograms of Alexa$_{647}$ fluorescence for BBS-Q1-YFP co-expressed with either nanobody (black line) or nanoCHIP (red line). Plot generated from population of CFP-positive cells. Dotted line is threshold value for Alexa$_{647}$ signal. (**H,I**) Flow cytometry analyses of surface Q1 channels in cells expressing BBS-Q1 with either nanobody alone or with nanoCHIP. Same format as (**F,G**).

DOI: https://doi.org/10.7554/eLife.29744.004

The following figure supplements are available for figure 2:

**Figure supplement 1.** Titration of nanoCHIP expression.
DOI: https://doi.org/10.7554/eLife.29744.005

**Figure supplement 2.** Catalytically inactive nanoCHIP* has no effect on Q1 surface expression.
DOI: https://doi.org/10.7554/eLife.29744.006

**Figure supplement 3.** Screening the effects of different engineered E3 ligases.
DOI: https://doi.org/10.7554/eLife.29744.007

*Figure 2 continued on next page*

*Figure 2 continued*

**Figure supplement 4.** Engineered nanoNEDD4-2 decreases both Q1 surface density and total protein expression.
DOI: https://doi.org/10.7554/eLife.29744.008

(*Aromolaran et al., 2014*). The C-terminal YFP tag provides a fluorescent measure of total Q1 expression (*Figure 2A*). The nanoCHIP construct was generated in a P2A vector that expressed CFP as a separate reporter protein in a 1:1 ratio with nanoCHIP (*Figure 2B*). We performed two types of control experiments. First, BBS-Q1-YFP was expressed with nanobody-P2A-CFP alone (nano) (*Figure 2A*). Second, nanoCHIP was co-expressed with BBS-Q1 lacking the C-terminus YFP tag (*Figure 2C*). We used flow cytometry to rapidly quantify total (YFP) and surface (red; $BTX_{647}$) Q1 expression in ~50,000 live cells, with single cell resolution. Control cells (nano + BBS-Q1-YFP) displayed robust total Q1 expression (YFP signal) in CFP-positive cells (*Figure 2D*). Test cells expressing nanoCHIP + BBS-Q1-YFP showed little change in YFP fluorescence compared to control (YFP = 577 ± 11 a.f.u, *n* = 1727 for nano; YFP = 630 ± 18 a.f.u, *n* = 783 for nanoCHIP), suggesting that the presumed targeted ubiquitination did not substantively affect Q1 stability (*Figure 2D,E*). By contrast, the surface density of BBS-Q1-YFP between the two conditions revealed an entirely different picture. Whereas control cells (nano) displayed a sizable population of surface BBS-Q1-YFP as reported by robust mean red fluorescence signal ($BTX_{647}$ = 822 ± 26 a.f.u, *n* = 4837), this surface pool was almost completely eliminated in cells co-expressing nanoCHIP ($BTX_{647}$ = 55 ± 2 a.f.u, *n* = 2257) (*Figure 2F,G*). To assess specificity, we co-expressed nanoCHIP with BBS-Q1. In sharp contrast to the result obtained with BBS-Q1-YFP, nanoCHIP had no effect on surface expression of BBS-Q1 channels ($BTX_{647}$ = 527 ± 16 a.f.u, *n* = 6425 for nano; $BTX_{647}$ = 633 ± 24 a.f.u, *n* = 3657 for nanoCHIP) (*Figure 2H,I*). These data were obtained with a 1:3 transfection ratio of Q1 to nanoCHIP cDNA. Similar results were obtained using transfection ratios of 1:1 and 1:5 (*Figure 2—figure supplement 1*). As a further control, co-expression of a CHIP deletion mutant (nanoCHIP*) that abolishes E3 ligase activity (*Nikolay et al., 2004*) did not alter Q1 surface expression, confirming the requirement of catalytic activity for nanoCHIP-dependent surface modulation (*Figure 2—figure supplement 2*). We used a similar strategy to selectively target distinct E3 ligase classes to Q1-YFP, notably two members of the RING family (nanoNSlmb and nanoMDM2), as well as NEDD4-2 from the HECT family (nanoNEDD4-2) (*Figure 2—figure supplements 3* and *4*). We obtained qualitatively similar results with nanoNSlmb and nanoMDM2 to what we observed with nanoCHIP in that they all reduced BBS-Q1-YFP surface density with minimal effects on total expression (*Figure 2—figure supplement 3*). Interestingly, nanoNEDD4-2 diminished total Q1-YFP expression concomitant with the decreased surface expression (*Figure 2—figure supplement 4*). Both effects on Q1 stability and surface density were abolished with co-expression of a NEDD4-2 catalytic inactive mutant (nanoNEDD4-2*) (*Figure 2—figure supplement 4*). Given that nanoCHIP had the most robust effect on reducing Q1 surface density (*Figure 2—figure supplement 3*) we focused the rest of the study on this engineered E3 ligase.

## nanoCHIP increases ubiquitination of Q1

Q1 is known to be ubiquitinated and regulated by heterologously expressed wild-type NEDD4-2 (*Jespersen et al., 2007*). As a prelude to determining whether nanoCHIP enhances ubiquitination of Q1-YFP, we first sought to reproduce the previously reported NEDD4-2-mediated ubiquitination of Q1 (*Jespersen et al., 2007*). We transiently expressed Q1-YFP either alone (control) or with NEDD4-2 in HEK293 cells. The cells were lysed under denaturing conditions and Q1-YFP was pulled down with anti-Q1 antibody. Western blot of the immunoprecipitated Q1-YFP using anti-Q1 displayed four bands representing the monomeric, dimeric, trimeric, and tetrameric channel species (*Figure 3A*). Densitometric analyses of the bands indicated that co-expression with NEDD4-2 reduced the total expression of Q1-YFP (*Figure 3A*; area under the curve), in agreement with previous results (*Jespersen et al., 2007*). Flow cytometry measurements were also consistent with this result (*Figure 3—figure supplement 1*). Having confirmed Q1-YFP pulldown, the membrane was stripped and probed with anti-ubiquitin (*Figure 3B*). Control cells expressing Q1-YFP alone displayed some ubiquitination reflecting the activity of an endogenous E3 ligase(s). Co-expression of

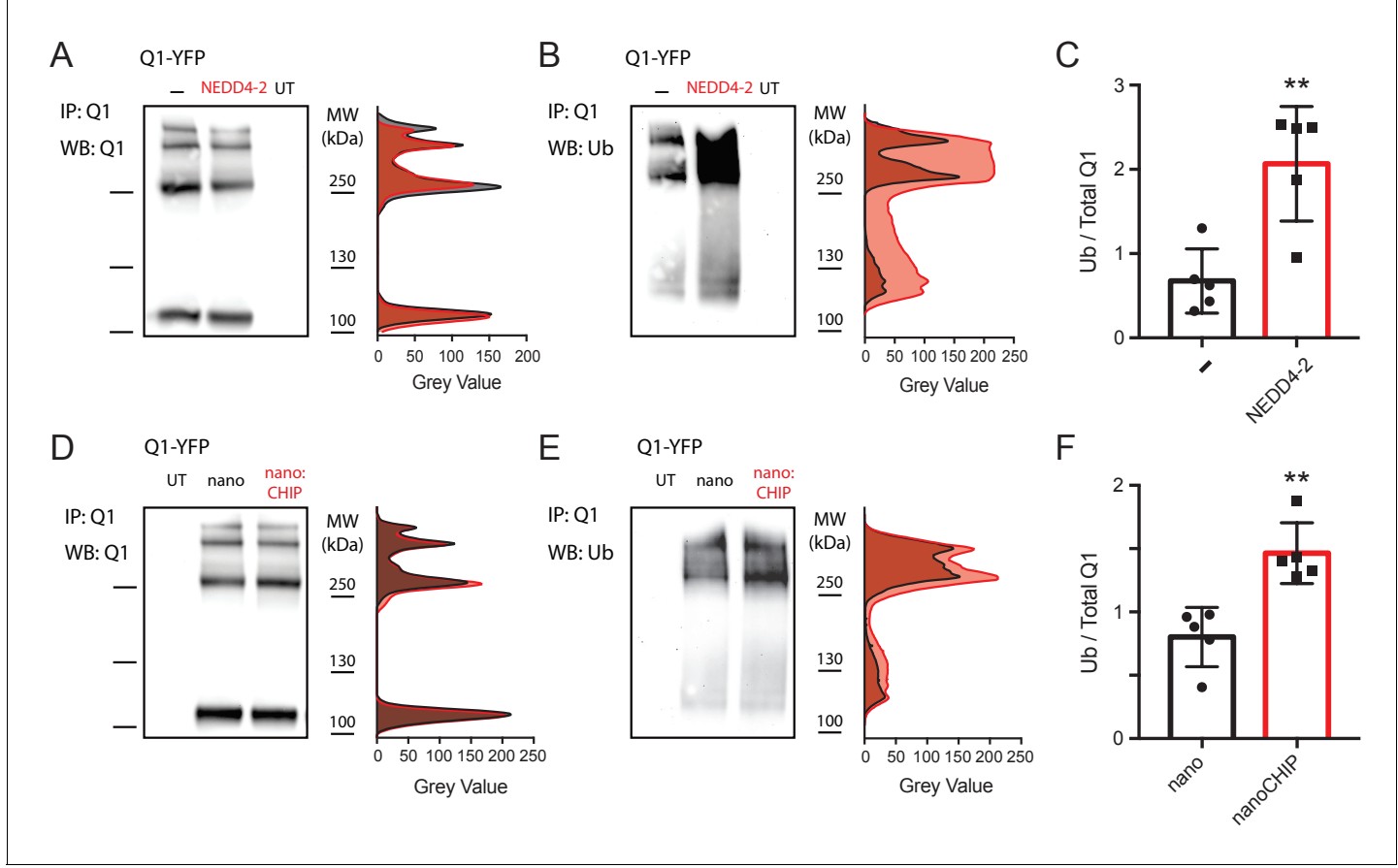

**Figure 3.** nanoCHIP increases Q1 ubiquitination. (**A**) *Left*, Q1 pulldowns probed with anti-Q1 antibody from HEK293 cells expressing Q1-YFP ± NEDD4-2, and untransfected controls (UT). *Right*, densitometric analyses of anti-Q1 Western blot bands. (**B**) *Left*, Anti-ubiquitin labeling of the stripped Western blot from (**A**). *Right*, densitometric analyses of anti-ubiquitin Western blot bands. (**C**) Relative Q1 ubiquitination computed by ratio of anti-ubiquitin to anti-Q1 signal intensity. (**D–F**) Anti-Q1 and anti-ubiquitin Western blot signals from HEK293 cells expressing Q1-YFP with either nanobody alone or nanoCHIP. Same format as (**A–C**). **p<0.01, Student's *t* test.
DOI: https://doi.org/10.7554/eLife.29744.009

The following figure supplement is available for figure 3:

**Figure supplement 1.** Full-length NEDD4-2 diminishes both Q1 surface density and total protein expression.
DOI: https://doi.org/10.7554/eLife.29744.010

NEDD4-2 substantially increased Q1-YFP ubiquitination compared to the control condition (*Figure 3B,C*).

With the method for detecting Q1-YFP ubiquitination validated, we turned to the effect of nano-CHIP in this biochemical assay. Consistent with the flow cytometry experiments, nanoCHIP did not substantively affect Q1-YFP total expression (*Figure 3D*). Nevertheless, nanoCHIP significantly augmented Q1-YFP ubiquitination compared to the control condition, although the effect was smaller than observed with NEDD4-2 (*Figure 3E,F*). As such, our findings suggest that modest changes in total ubiquitination intensity as detected by conventional Western blot can result in substantial functional and cell biological effects on ion channel surface trafficking.

## nanoCHIP regulation of Q1/KCNE1 complexes

Physiologically, Q1 is typically associated with auxiliary KCNE subunits that are single transmembrane spanning proteins. There are five distinct KCNE subunits (KCNE1-KCNE5), each of which can profoundly shape the outward $K^+$ current waveform when associated with Q1 (*Sun et al., 2012*). The interaction of KCNE1 with Q1 transforms the current waveform from one which is small and rapidly activating (Q1 alone) to one that is large and slowly activating ($I_{Ks}$; Q1 + KCNE1) (*Barhanin et al.,*

*1996*; *Sanguinetti et al., 1996*). The slowly activating kinetics of $I_{Ks}$ is crucial to its physiological role in human cardiac action potential repolarization.

We determined whether nanoCHIP could regulate the Q1/KCNE1 macromolecular complex. Similar to our observations with Q1 alone, nanoCHIP dramatically reduced surface density of BBS-Q1-YFP co-expressed with KCNE1 (*Figure 4A–C*), while minimally affecting total protein expression (*Figure 4—figure supplement 1*). This effect was selective, as in cells expressing BBS-Q1 + KCNE1 (lacking a YFP tag), nanoCHIP had no effect on channel surface density (*Figure 4D–F*). Finally, we wondered whether we could manipulate the surface expression of Q1 by targeting nanoCHIP to the auxiliary KCNE1 subunit. Indeed, in cells expressing BBS-Q1 + KCNE1-YFP, nanoCHIP effectively and selectively eliminated surface channels (*Figure 4G–I* and *Figure 4—figure supplement 1*), demonstrating the potential power of the approach to sculpt ion channel macromolecular complexes by targeting accessory proteins. Consistent with these results, nanoCHIP targeted to KCNE1-YFP markedly increased ubiquitination of co-expressed Q1 (*Figure 4J,K*).

We next examined the impact of nanoCHIP on functional $I_{Ks}$ currents, under conditions that mirrored those used to examine channel trafficking and total Q1 expression. Control Chinese hamster ovary (CHO) cells transfected with Q1-YFP + KCNE1 + nano displayed robust outward $K^+$ currents with the signature slow activation kinetics of $I_{Ks}$, which were essentially eliminated by nanoCHIP (*Figure 5A–C*). A similar result was observed in HEK293 cells (*Figure 5—figure supplement 1*). By contrast, nanoCHIP had no effect on $I_{Ks}$ reconstituted with subunits that lacked the YFP tag (*Figure 5D–F*). Finally, switching the YFP tag to KCNE1 also yielded $I_{Ks}$ that was significantly reduced by nanoCHIP (*Figure 5G–I*).

## An inducible system for temporal control of Q1 ubiquitination and trafficking

We next sought to exploit the rapamycin-induced heterodimerization system to develop an approach that enables acute temporal control of ubiquitination of specific target proteins (*Crabtree and Schreiber, 1996*). We separated the nanoCHIP catalytic (CHIP) and substrate-binding (nano) domains, and fused them to the rapamycin binding proteins FRB and FKBP, respectively (*Inoue et al., 2005*; *Yang et al., 2007*) (*Figure 6A*). FRB-CHIP and FKBP-nano are expected to have a low affinity for each other under basal conditions. Hence, when these two constructs are co-expressed with BBS-Q1-YFP we would not expect any channel ubiquitination by the engineered FRB-CHIP (*Figure 6A*). Application of the small molecule rapamycin would then facilitate FRB-FKBP interaction, effectively recruiting the CHIP catalytic domain to BBS-Q1-YFP and initiating ubiquitination (*Figure 6A*). We tested the effectiveness of this inducible nanoCHIP (iN-CHIP) approach by measuring the kinetics of rapamycin-induced decrease in surface density of BBS-Q1-YFP (*Figure 6B*). Utilizing the flow cytometry-based fluorescence assay, we observed a time-dependent decrease in the surface pool of BBS-Q1-YFP after adding rapamycin. A measurable effect was seen within 20 min of rapamycin addition to the transfected HEK293 cells; the half-life for reduction of surface channels was 209 ± 17 mins (*Figure 6B,C*). In control experiments, rapamycin treatment for 20 hr had no impact on Q1 surface density in cells expressing BBS-Q1 together with FRB-CHIP and FKBP-nano (*Figure 6—figure supplement 1*).

In principle, the nanoCHIP-mediated reduction in BBS-Q1-YFP surface density could be due to: a decreased rate of delivery of new Q1 channels to the surface; an increased rate of internalization (or removal) of Q1 channels from the cell surface; or a combination of the two processes. To distinguish among these possibilities, we employed iN-CHIP along with two optical pulse-chase methods to measure the rates of BBS-Q1-YFP delivery to and removal from the cell surface (*Figure 7*).

HEK293 cells were transiently transfected with BBS-Q1-YFP + FRB-CHIP + FKBP-nano. To measure rate of BBS-Q1-YFP delivery to the cell surface, we incubated live, non-permeabilized cells at 4°C to halt all trafficking processes, and exposed them to unconjugated BTX to block all extracellular BBS epitopes initially present at the plasma membrane (pulse). Cells were then incubated at 37°C for varying epochs during which trafficking processes resumed, including delivery of new BBS-tagged channels to the surface (chase). Cells were then returned to 4°C and the newly delivered surface channels labeled with $BTX_{647}$ and quantified by flow cytometry (*Figure 7A*). When this experiment was conducted in the absence of rapamycin pre-treatment, we observed robust delivery of new BBS-Q1-YFP to the cell surface during the chase period with a half-life of ~36 ± 3 mins (*Figure 7B*). When cells were pretreated with rapamycin for 3 hr, delivery of new channels to the surface

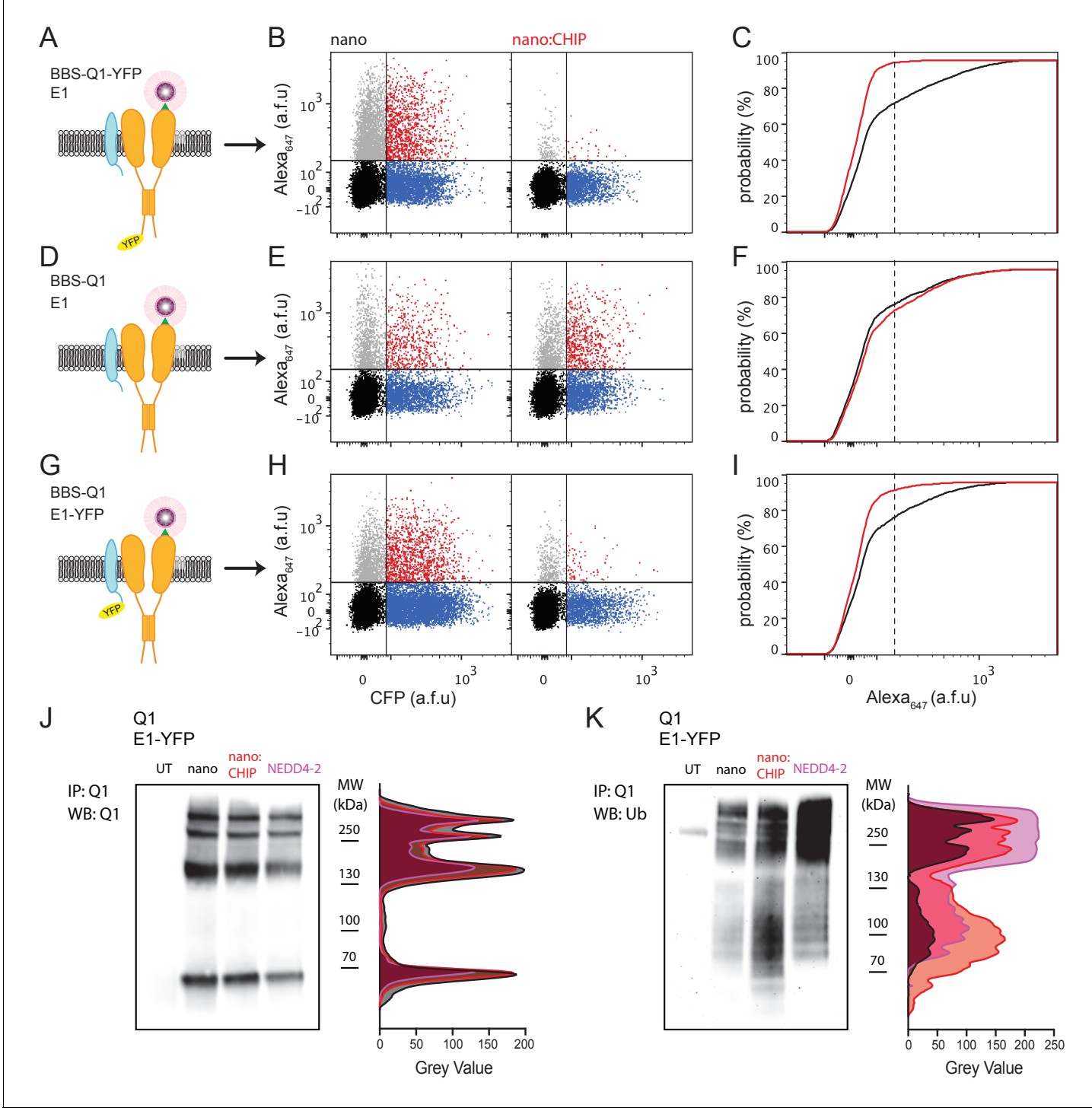

**Figure 4.** nanoCHIP regulation of Q1/KCNE1 complex surface density. (**A**) Cartoon of BBS-Q1-YFP + KCNE1. (**A**) Flow cytometry analyses of surface Q1 channels (Alexa$_{647}$ fluorescence) in HEK293 cells expressing BBS-Q1-YFP + KCNE1 with either nanobody (*left*, control) or nanoCHIP (*right*). (**C**) Cumulative distribution histograms of Alexa$_{647}$ fluorescence for BBS-Q1-YFP co-expressed with either nanobody (black line) or nanoCHIP (red line). (**D–F**) Schematic, flow cytometry analyses, and cumulative distribution histograms from cells expressing BBS-Q1 + KCNE1 with either nanobody alone or nanoCHIP. Same format as (**A–C**). (**G–I**) Schematic, flow cytometry analyses, and cumulative distribution histograms from cells expressing BBS-Q1 + KCNE1-YFP with either nanobody alone or nanoCHIP. Same format as (**A–C**). (**J**) *Left*, Q1 pulldowns probed with anti-Q1 antibody from HEK293 cells expressing Q1 + KCNE1-YFP with nanobody alone, nanoCHIP, or NEDD4-2. UT, untransfected cells. *Right*, densitometric analyses of anti-Q1 Western blot bands for the different conditions. (**K**) *Left*, Anti-ubiquitin labeling of the stripped Western blot from (**J**). *Right*, densitometric analyses of anti-ubiquitin Western blot bands.

*Figure 4 continued on next page*

*Figure 4 continued*

DOI: https://doi.org/10.7554/eLife.29744.011

The following figure supplement is available for figure 4:

**Figure supplement 1.** nanoCHIP has no effect on Q1/E1 total protein expression.

DOI: https://doi.org/10.7554/eLife.29744.012

plateaued at a value ~75% lower compared to control (*Figure 7B*). Thus, iN-CHIP-induced ubiquiti-nation of BBS-Q1-YFP compromises forward trafficking of channels to the cell surface.

To evaluate channel removal from the surface, we labeled live, non-permeabilized HEK293 cells (expressing BBS-Q1-YFP + FRB-CHIP + FKBP-nano) with biotinylated bungarotoxin (BTX-biotin) at 4°C (pulse). Cells were then incubated at 37°C for varying time periods to resume trafficking

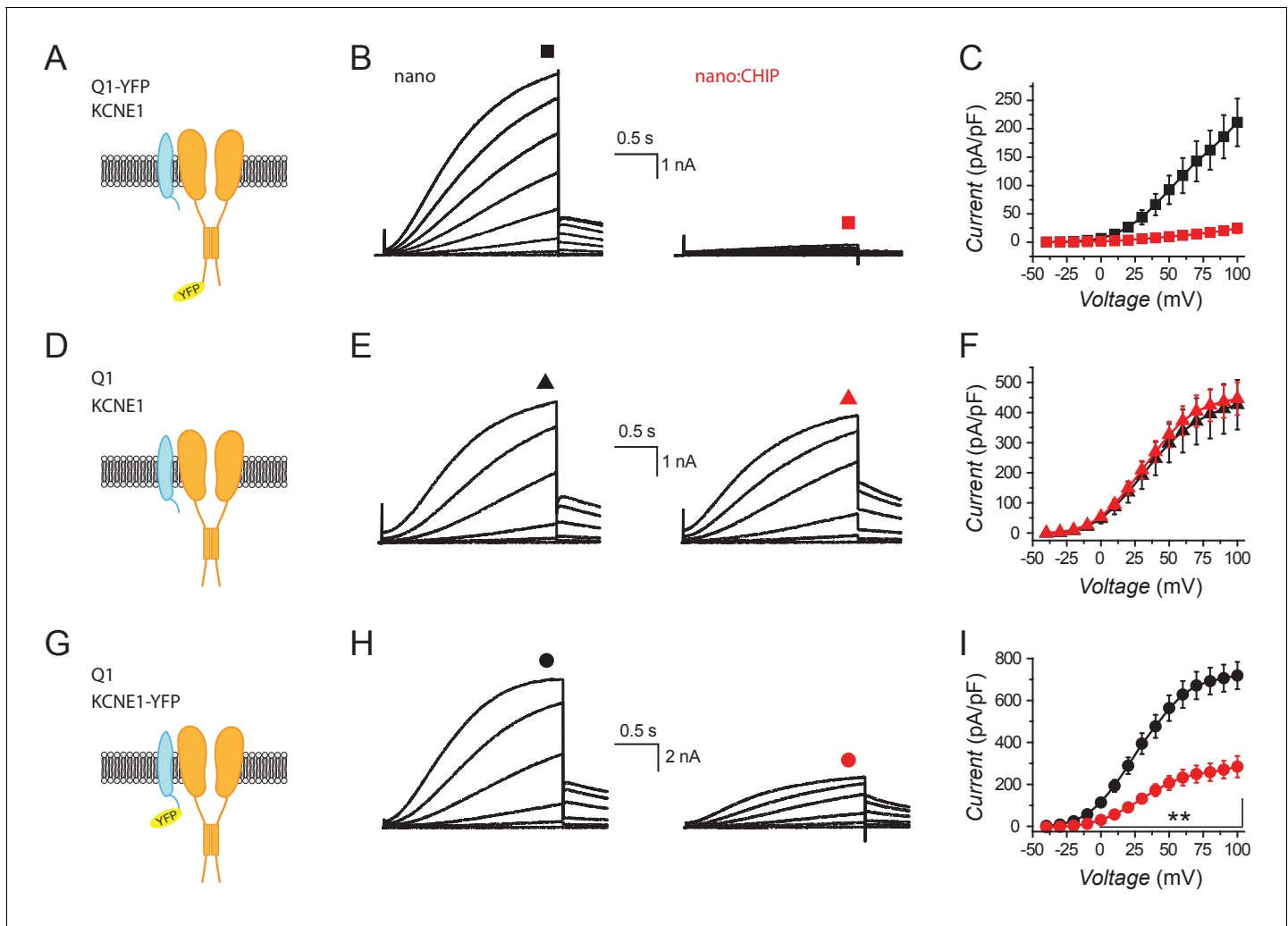

**Figure 5.** Functional knockdown of reconstituted $I_{ks}$ by nanoCHIP. (**A**) Schematic, Q1-YFP + KCNE1. (**B**) Exemplar family of $I_{Ks}$ reconstituted in CHO cells expressing Q1-YFP + KCNE1 with either nanobody alone (*left*) or nanoCHIP (*right*). (**C**) Population I-V curves for nano (■, *n* = 5) and nanoCHIP (■, *n* = 5). (**D–F**) Schematic, exemplar currents and population I-V curves for CHO cells expressing Q1 + KCNE1 with either nanobody (▲, *n* = 14) or nanoCHIP (▲, *n* = 12). Same format as (**A–C**). (**G–I**) Schematic, exemplar currents and population I-V curves for CHO cells expressing Q1 + KCNE1-YFP with either nanobody (●, *n* = 13) or nanoCHIP (●, *n* = 8). Same format as (**A–C**). **p<0.01, Student's *t* test.

DOI: https://doi.org/10.7554/eLife.29744.013

The following figure supplement is available for figure 5:

**Figure supplement 1.** Functional knockdown of $I_{ks}$ reconstituted in HEK293 cells by nanoCHIP.

DOI: https://doi.org/10.7554/eLife.29744.014

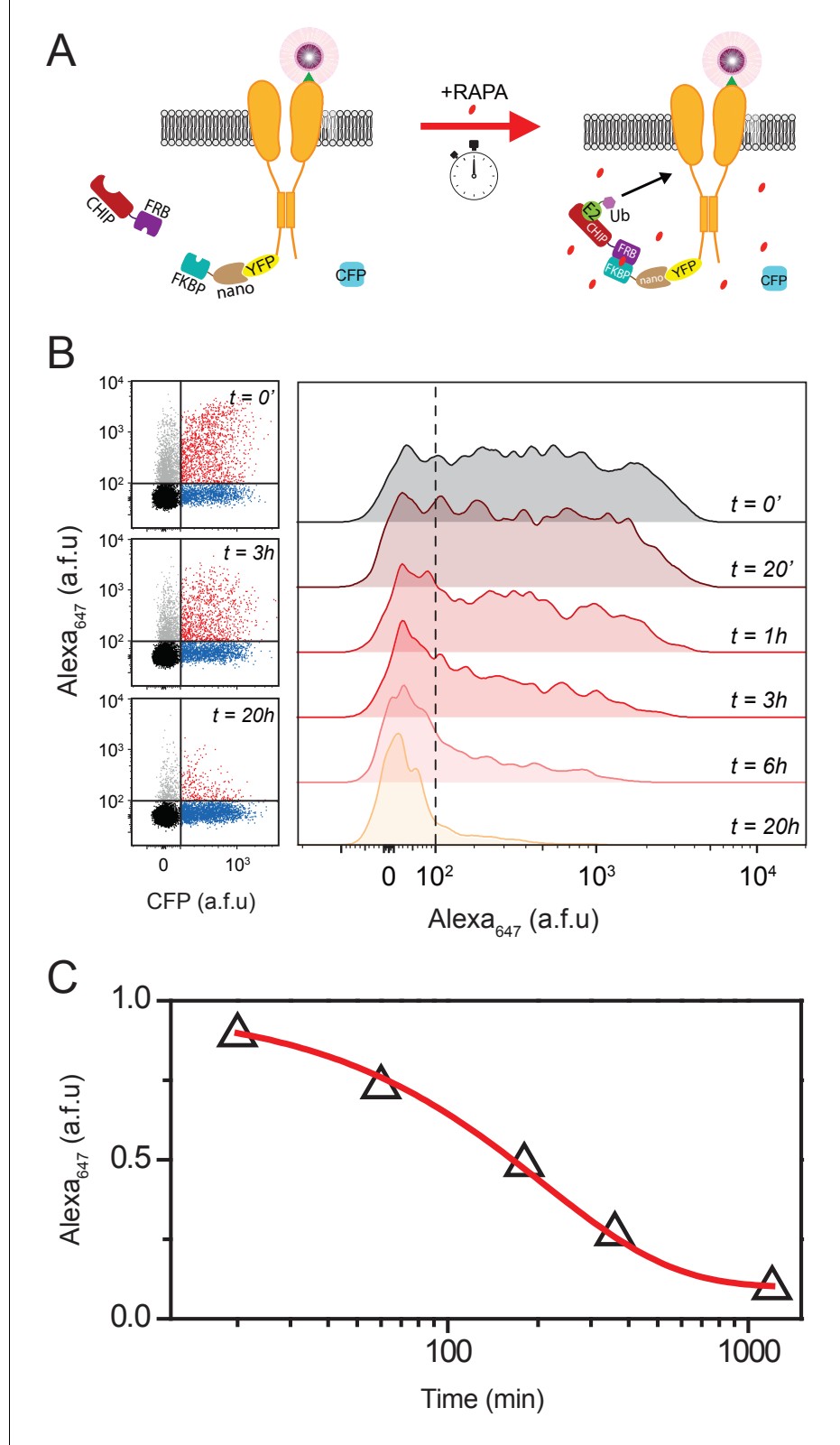

**Figure 6.** A small-molecule-inducible system for temporal control of Q1 surface expression. (**A**) Cartoon showing FKBP/FRB heterodimerization strategy for rapamycin-induced recruitment of engineered E3 ligase (iN-CHIP) to Q1-YFP. (**B**) Representative flow cytometry dot plots (left) and histograms (right) showing evolution of surface Q1 channels in cells expressing BBS-Q1-YFP + FRB-CHIP + FKBP-nano at varying time intervals after rapamycin

*Figure 6 continued on next page*

*Figure 6 continued*

induction. (C) Plot of normalized mean Q1 surface density (Alexa$_{647}$ fluorescence) as a function of time after rapamycin induction ($\Delta$, *n* = 1355–1523 cells; *N* = 3). Smooth curve is an exponential decay function fit to the data: $y = Ae^{\frac{-t}{\tau}} + y0$, with A = 0.87 ± 0.03, y0 = 0.10 ± 0.02, $\tau$ = 208.8 ± 17.4 mins.

DOI: https://doi.org/10.7554/eLife.29744.015

The following figure supplement is available for figure 6:

**Figure supplement 1.** . iN-CHIP does not modulate surface expression of untagged BBS-Q1.

DOI: https://doi.org/10.7554/eLife.29744.016

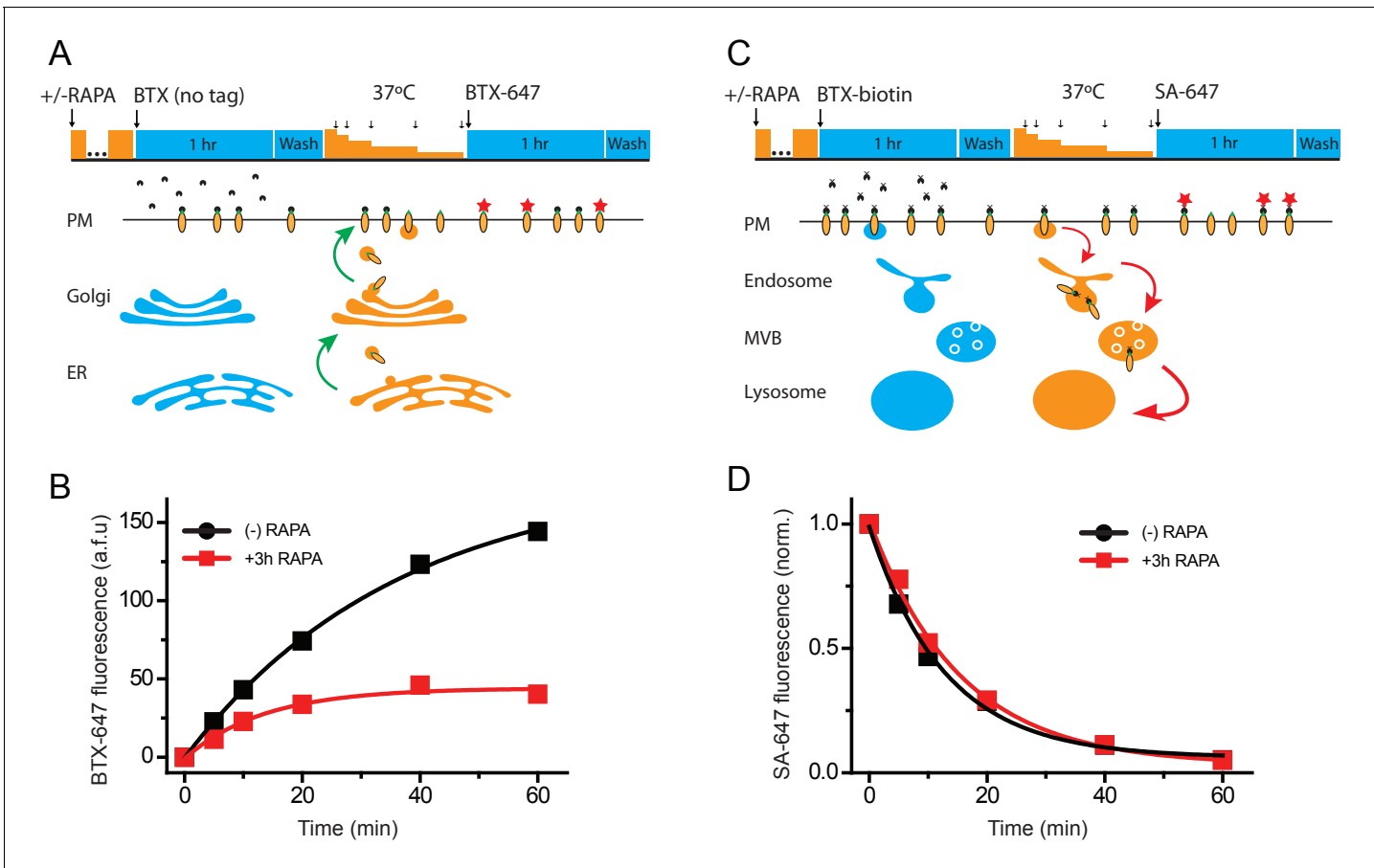

**Figure 7.** iN-CHIP selectively impairs Q1-YFP forward trafficking. (**A**) Schematic showing optical pulse-chase assay for measuring BBS-Q1-YFP forward trafficking. Cells expressing BBS-Q1-YFP + FRB-CHIP + FKBP-nano were induced with rapamycin and initial surface channels blocked by incubation with untagged α-bungarotoxin (BTX) at 4°C. Cells were washed and placed back at 37°C for varying time intervals (5, 10, 20, 40, 60 min) to resume delivery of new channels to the surface membrane. Newly delivered channels were labeled with Alexa Fluor 647 conjugated BTX (BTX$_{647}$) at 4°C and analyzed using flow cytometry. (**B**) Time evolution of BBS-Q1-YFP delivery to the surface without (•, *n* = 2878–3905 cells; *N* = 2) or with (■, *n* = 2990–3469 cells; *N* = 2) rapamycin induction. Smooth curves are fits of an exponential growth function to the data: $y = Ae^{\frac{-t}{\tau}} + y0$. For •; A = −45.1 ± 3.8, y0 = 44.1 ± 3.0, $\tau$ = 13.4 ± 3.0 mins. For ■; A = −180.5 ± 7.0, y0 = 179.9 ± 7.7, $\tau$ = 36.1 ± 3.2 mins. (**C**) Schematic showing optical assay for measuring BBS-Q1-YFP internalization. Cells expressing BBS-Q1-YFP + FRB-CHIP + FKBP-nano were induced with rapamycin and initial surface channels labeled with biotin-conjugated BTX (BTX-biotin) at 4°C. Cells were washed and incubated at 37°C for varying time intervals (5, 10, 20, 40, 60 min) to allow for internalization of surface channels. The remaining surface channels were labeled with Alexa Fluor 647-conjugated streptavidin (SA-647) at 4°C. (**D**) Time evolution of loss of surface BBS-Q1-YFP channels without (•, *n* = 3430–4919 cells; *N* = 2) or with (■, n = 3336–4744 cells; *N* = 2) rapamycin induction. Smooth curves are fits of an exponential decay function to the data: $y = Ae^{\frac{-t}{\tau}} + y0$. For •; A = 0.93 ± 0.03, y0 = 0.06 ± 0.02, $\tau$ = 12.8 ± 1.1 mins. For ■; A = 0.98 ± 0.03, y0 = 0.03 ± 0.03, $\tau$ = 15.3 ± 1.3 mins.

DOI: https://doi.org/10.7554/eLife.29744.017

processes (chase). Following the chase period, cells were labeled with streptavidin-conjugated Alexa Fluor 647 at 4°C. In this paradigm, red fluorescent labeling would only occur on channels that were initially present at the surface and labeled with BTX-biotin during the pulse period. A decrease in fluorescence with increasing chase times would be expected due to internalization of BTX-biotin-labeled channels (*Figure 7C*). Indeed, control cells (no rapamycin pre-treatment) displayed an exponential decline in red fluorescence with increasing chase time (*Figure 7D*). Surprisingly, pre-activation of iN-CHIP with a 3 hr rapamycin pre-treatment had no impact on the rate of BBS-Q1-YFP internalization (*Figure 7D*). Together, the results indicate that nanoCHIP reduces BBS-Q1-YFP surface density by selectively reducing forward trafficking of the channel.

## Impact of nanoCHIP on Q1 expressed in adult cardiomyocytes

Ultimately, the general usefulness of the engineered E3 ligase approach to manipulate functional expression of membrane proteins hinges critically on the system performing robustly in native cells and tissues which have a more complex intracellular environment compared to heterologous cells. We tested the ability of nanoCHIP to suppress the surface density of BBS-Q1-YFP expressed in adult rat ventricular myocytes. We generated adenoviral vectors for BBS-Q1-YFP, nano-P2A-CFP, and nanoCHIP-P2A-CFP and used these to infect cultured cardiomyocytes. Control cells expressing BBS-Q1-YFP + nano displayed strong YFP/CFP fluorescence as well as $QD_{655}$ signal on the sarcolemma,

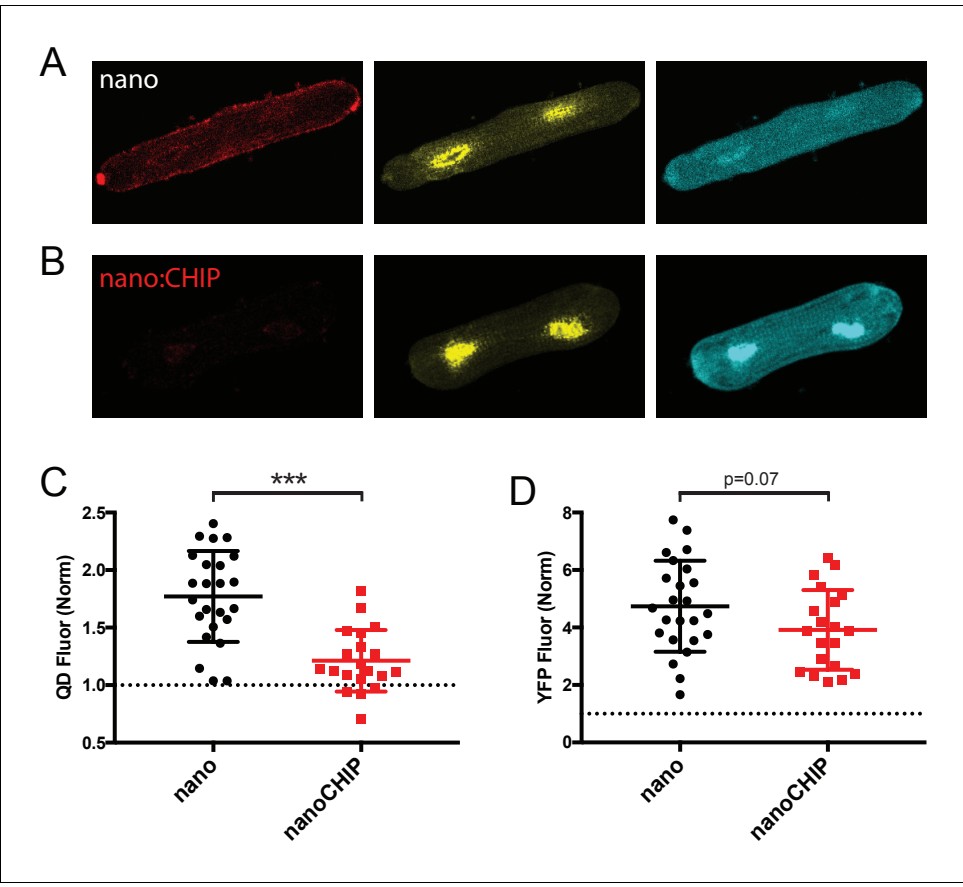

**Figure 8.** nanoCHIP eliminates Q1 surface expression in cardiomyocytes. (A) Exemplar adult rat cardiomyocyte expressing BBS-Q1-YFP + nanobody-P2a-CFP. Fluorescence shows surface Q1 ($QD_{655}$ signal, *left*), total Q1 (YFP signal, *middle*) and marker for nanobody expression (CFP signal, *right*). (B) Exemplar cardiomyocyte co-expressing BBS-Q1-YFP and nanoCHIP. Same format as (A). (C) Comparison of surface Q1 channels in cardiomyocytes co-expressing either nanobody (•, *n* = 24) or nanoCHIP (■, *n* = 20). ***p<0.0001, Student's unpaired *t* test. (D) Comparison of total Q1 channels in cardiomyocytes co-expressing either nanobody (•, *n* = 24) or nanoCHIP (■, *n* = 20).
DOI: https://doi.org/10.7554/eLife.29744.018

indicating robust cell surface density of the channel (*Figure 8A,C*). By contrast, cardiomyocytes co-expressing nanoCHIP showed a sharply depressed $QD_{655}$ signal (*Figure 8B,C*). YFP fluorescence was not significantly different between the two experimental conditions (*Figure 8D*). These data demonstrate that nanoCHIP is effective in cardiomyocytes, and selectively down-regulates BBS-Q1-YFP surface pool in this native cellular context.

## nanoCHIP selectively down-regulates surface $Ca_V1.2$ channels

To test the generalizability of the engineered E3 ligase approach for regulating ion channels, we probed whether nanoCHIP could modulate the trafficking of a recombinant voltage-gated calcium channel ($Ca_V1.2$). Cav1.2 mediates excitation-contraction coupling and excitation-transcription coupling in heart and neurons, respectively, and is comprised of a pore-forming $\alpha_{1C}$ and accessory ($\beta$, $\alpha_2\delta$, $\gamma$) subunits (*Catterall, 2000*). We attached YFP to the C-terminus of $\alpha_{1C}$ to render it a putative substrate for nanoCHIP, and a BBS epitope tag on an extracellular loop to enable fluorescent detection of surface channels (*Figure 9A*) (*Yang et al., 2010*; *Subramanyam et al., 2013*). Similar to our observations with Q1, nanoCHIP selectively eliminated the surface $Ca_V1.2$ pool with no impact on total BBS-$\alpha_{1C}$-YFP expression (*Figure 9B–E*). Consistent with these results, nanoCHIP essentially abolished $Ca_V1.2$ currents reconstituted in HEK 293 cells (*Figure 9F,G*).

## Discussion

In this work, we have developed a toolset that enables post-translational ubiquitination of proteins in a specific and controllable manner, and applied these to study ubiquitin regulation of two distinct ion channels— KCNQ1 and $Ca_V1.2$. The approach has two major utilities: (1) it provides a method to facilitate mechanistic understanding of how ubiquitination may regulate diverse aspects of membrane protein fate including, trafficking, stability, and function; and (2) it provides a method for regulating functional expression of ion channel macromolecular complexes in a manner that complements, and has particular advantages over, pre-existing genomic/mRNA interference technologies. We discuss these two aspects of the work in the context of previous studies.

### Complexities in decoding ubiquitin regulation of membrane proteins

The precise mechanisms and signals regulating the dynamic trafficking of ion channels among membrane compartments are not completely understood and difficult to study, in part, due to their complexity and a lack of enabling tools. This is a serious limitation given that a number of ion channelopathies (e.g. cystic fibrosis, epilepsy, Liddle syndrome, cardiac arrhythmias) may arise due to dysregulation in ion channel surface expression (*Abriel and Staub, 2005*).

Ubiquitination is a critical post-translational modification capable of regulating diverse aspects of protein fate including trafficking, sorting, and stability. Several ion channels and transporters are known to be regulated by ubiquitination, including ENaC, ClC-5, KCNQ1, and $Na_V$ channels, which have all been found to be regulated by NEDD4-like family ubiquitin ligases (*Staub et al., 1997*; *Abriel et al., 1999*; *Schwake et al., 2001*; *Fotia et al., 2004*; *van Bemmelen et al., 2004*; *Abriel and Staub, 2005*; *Jespersen et al., 2007*). Many of these channels possess PY motifs (P-P-X-Y-X-X-$\phi$ where P is proline, Y is tyrosine, X is any amino acid, and $\phi$ is a hydrophobic residue) to which NEDD4-like proteins bind using WW protein interaction modules. In co-expression studies, NEDD4-like proteins have been shown to bind to these membrane proteins, to increase their ubiquitination, and to promote their degradation presumably via targeting to lysosomes (*Staub et al., 1996*; *Staub et al., 1997*; *Abriel et al., 1999*). Nevertheless, significant questions remain regarding underlying mechanisms due to complexities in the ubiquitin regulatory system that arises at three levels. First, multiple E3 ligases may converge to regulate a single substrate. Second, a single E3 ligase such as NEDD4-2 catalyzes ubiquitination of multiple substrates. Third, distinct E3 ligases can give rise to monoubiquitination or polyubiquitin chains with differing lysine chain linkages and, thus, divergent functional consequences (*Abriel and Staub, 2005*; *Komander, 2009*; *MacGurn et al., 2012*; *Foot et al., 2017*).

The method of specifically targeting E3 catalytic domains to selected membrane proteins offers opportunities to dissect these inherent intricacies of the ubiquitin regulatory system. Using this approach, our results yield new insights into ubiquitin regulation of Q1 and $Ca_V1.2$ channels. First, we found that nanoCHIP increased ubiquitination of both YFP-tagged Q1 and $Ca_V1.2$ without

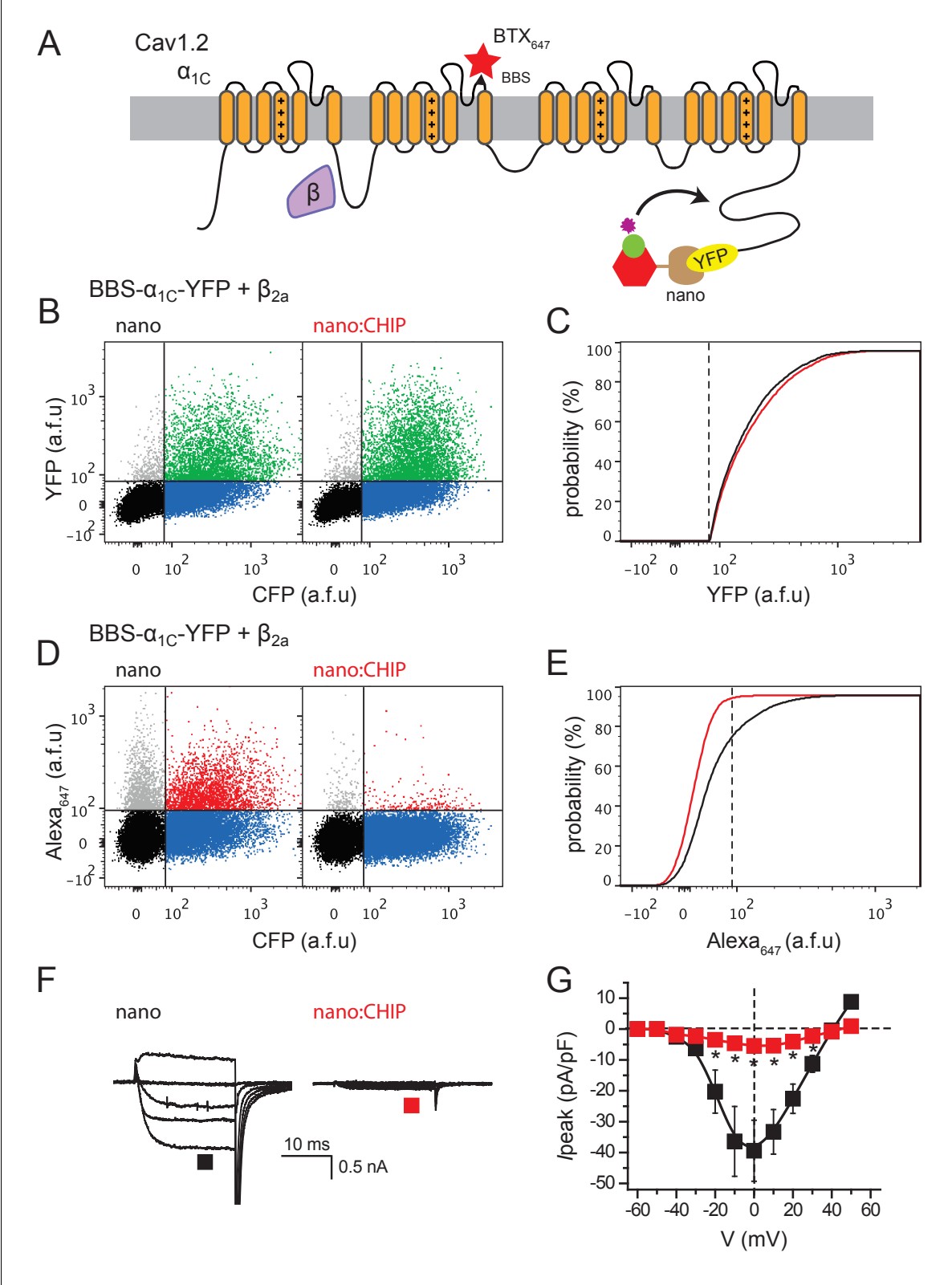

**Figure 9.** Functional knockdown of recombinant Cav1.2 channels by nanoCHIP. (**A**) Schematic of Ca_V1.2 pore-forming $\alpha_{1C}$ and auxiliary $\beta$ subunit complex. An extracellular BBS epitope tag placed on the domain II S5-S6 loop permits detection of surface channels using Alex Fluor647-conjugated bungarotoxin. A YFP tag on $\alpha_{1C}$ C-terminus reports on the total expression and serves as a docking site for nanoCHIP. (**B**) Flow cytometry analyses of total $\alpha_{1C}$ expression (YFP fluorescence) in cells expressing BBS-$\alpha_{1C}$-YFP + $\beta_{2a}$ together with either nanobody (*left*, control) or nanoCHIP (*right*). (**C**)

*Figure 9 continued on next page*

*Figure 9 continued*

Cumulative distribution histograms of YFP fluorescence for BBS-$\alpha_{1C}$-YFP + $\beta_{2a}$ co-expressed with either nanobody (black line) or nanoCHIP (red line). (D) Flow cytometry analyses of surface $\alpha_{1C}$ (Alexa$_{647}$ fluorescence) in cells expressing BBS-$\alpha_{1C}$-YFP + $\beta_{2a}$ + nanobody (*left*, control) or BBS-$\alpha_{1C}$-YFP + $\beta_{2a}$ + nanoCHIP (*right*). (E) Cumulative distribution histograms of Alexa$_{647}$ fluorescence for BBS-$\alpha_{1C}$-YFP + $\beta_{2a}$ co-expressed with either nanobody (black line) or nanoCHIP (red line). (F) Exemplar Ba$^{2+}$ currents from Ca$_V$1.2 channels reconstituted in HEK293 cells expressing $\alpha_{1C}$-YFP + $\beta_{2a}$ with either nanobody alone (*left*) or nanoCHIP (*right*). (G) Population I-V curves for $\alpha_{1C}$-YFP + $\beta_{2a}$ reconstituted with either nano (■, n = 10) or nanoCHIP (■, n = 14). *p<0.05, Student's unpaired *t* test.

DOI: https://doi.org/10.7554/eLife.29744.019

significantly impacting total protein expression, demonstrating that mere ubiquitination of these channels is not sufficient to direct their degradation. By contrast, NEDD4-2 both enhances ubiquitination and decreases stability of Q1 (*Jespersen et al., 2007*) (*Figure 3A* and *Figure 3—figure supplement 1*). A possible explanation for this difference could lie in the fact that NEDD4-like ubiquitin ligases catalyze ubiquitination of multiple protein substrates, including components of the ESCRT complex involved in sorting membrane proteins on endosomes into multi-vesicular bodies and lysosomes (*MacGurn et al., 2012*). Alternatively, the intrinsic differences in the type of ubiquitination conferred between NEDD4-2 and CHIP E3 ligase could be important. The finding that nanoNEDD4-2 was more effective than nanoCHIP in reducing Q1 stability, mirroring full-length NEDD4-2, is consistent with this interpretation. Overall, these results emphasize that distinct ligases can differentially impact the stability and subcellular localization of ion channels. Moreover, this work illustrates how engineered E3 ligases can be utilized to systematically and selectively probe the impact of particular E3 ligases on target proteins in the complex cellular environment. The targeted E3 ligase approach is complementary to a recently developed method that uses ubiquitin variants that are selective for distinct HECT ligases, and can either activate or inhibit their cognate E3 (*Zhang et al., 2016*).

A second unique observation was that iN-CHIP suppressed Q1 surface density by selectively diminishing forward trafficking of the channel, with no apparent effect on the rate of endocytosis. A common assumption is that ubiquitination of membrane proteins enhances their internalization from the cell surface, although recent results suggest a more complex picture in mammalian cells. For example, the E3 ligase Cbl is important to the internalization of activated epidermal growth factor receptors (EGFRs). Nevertheless, mutation of ubiquitination sites on EGFRs did not abolish their endocytosis (*Huang et al., 2007*). Similarly, a ubiquitin-deficient $\beta$2 adrenergic receptor was also internalized to a similar extent as the wild type protein (*Shenoy et al., 2001*). A complication in our experimental system is that Q1 displays a basal level of ubiquitination due to the activity of an endogenous E3 ligase of unknown identity. It is possible that this basal level of ubiquitination is sufficient to set a level of internalization that is not further enhanced by nanoCHIP.

The iN-CHIP-induced decrease in forward trafficking is intriguing though questions remain regarding the mechanistic bases of this effect. There are endogenous ubiquitin-mediated quality control mechanisms that would be expected to limit forward trafficking of membrane proteins. For example, the ER-associated degradation (ERAD) pathway is a prominent quality control mechanism which is accomplished by a chaperone-mediated ubiquitination of misfolded membrane proteins which are then retrotranslocated to the cytosol and targeted to the proteasome for degradation (*MacGurn et al., 2012*; *Foot et al., 2017*). There are also ubiquitin-dependent quality control mechanisms present at the Golgi which results in the diversion of membrane proteins to endosomes where they are sorted by the ESCRT system into multi-vesicular bodies and fusion with lysosomes (*Fire et al., 1998*; *MacGurn et al., 2012*; *Foot et al., 2017*). Both these quality control pathways result in the degradation of target proteins which is fundamentally different from our observations of the impact of nanoCHIP on Q1 and Ca$_V$1.2 channels. Ultimately, precise identification of the intracellular compartments in which nanoCHIP arrested Q1 and Ca$_V$1.2 channels reside will be important for deducing the mechanism of the compromised forward trafficking of these channels.

## Engineered E3 ligase approach as a tool to manipulate functional expression of membrane proteins

Eliminating protein function by preventing expression or with pharmacological agents is a cornerstone of modern biological research and disease therapy. Several approaches have been developed to eradicate expression of target proteins by interference at the genomic (knockout, zinc finger

nucleases, TALENs, CRISPR/Cas) or mRNA (siRNA, shRNA, microRNA) levels (*Fire et al., 1998*; *Gaj et al., 2013*; *Doudna and Charpentier, 2014*; *Boettcher and McManus, 2015*). These methods are widely used and powerful, but do have certain limitations that may be addressable with the tools developed here. The temporal control and resolution of mRNA interference methods is relatively poor because they are dependent on the degradation of the targeted native protein. For stable proteins with a long half-life this can adversely impact the efficacy of the mRNA interference approach. This gap can be potentially addressed by post-translational degradation of target proteins using engineered ubiquitin ligases. Indeed, several groups have utilized this approach to target diverse cytosolic proteins for degradation in situ. Yet none to date have applied this approach to ion channels, a specialized class of proteins that rely on a very different post-translational lifecycle of maturation, sorting, and trafficking.

The first implementation of targeted ubiquitination exploited SKP1-CUL1-F-box (SCF) E3 ligase complexes, in which a peptide motif was fused to a modular F-box-domain-containing protein in order to target the retinoblastoma tumor suppressor protein for degradation by a Cullin/RING E3 ligase complex (*Zhou et al., 2000*). Seeking a more generalizable approach, Caussinius et al fused the N-terminus F-box domain of Slmb (a *Drosophila melanogaster* F-box protein) to vhhGFP4 (NSlmb-vhhGFP4, also termed deGradFP), which effectively degraded various GFP-tagged proteins in *Drosophila* (*Caussinus et al., 2011*). Yet because approaches that exploit SCF complexes have the potential to sequester endogenous components by over-expressed engineered F-box proteins (*Hatakeyama et al., 2005*; *Portnoff et al., 2014*), other studies have utilized E3 ligases that are not similarly reliant on endogenous scaffold proteins for their mechanism of action. One such approach replaced the substrate binding (TPR) domain of CHIP to enable targeted ubiquitination and degradation of the proto-oncogenes c-Myc (*Hatakeyama et al., 2005*) and K-Ras (*Ma et al., 2013*). Further proof of concept experiments replaced the TPR of CHIP with intrabodies directed against β-galactosidase and maltose binding protein, which yielded effective destruction of these target proteins in transfected HEK293 cells (*Portnoff et al., 2014*). More recently, a technique referred to as Trim-Away exploits the high affinity of the E3 ligase TRIM21 for the Fc domain of antibodies for selective degradation of target proteins (*Clift et al., 2017*).

Against this background, nanoCHIP represents a hybrid of these previously engineered E3 ligases, wherein the TPR region of CHIP is replaced with a vhhGFP4 nanobody. Without exception, all previous renditions of this technology have relied on degradation of target proteins as the ultimate expression of efficacy. In comparison, our results suggest a fundamental difference between ion channels and cytosolic proteins in which targeted ubiquitination with nanoCHIP (as well as nanoNSlmb and nanoMDM2) yielded impaired trafficking and functional inactivation without frank degradation of the protein. As such, our results indicate that absolute degradation is not necessary for potent functional knockdown of Q1 and $Ca_V1.2$ channels, and highlight the importance of employing functional/cell biological assays to assess the efficacy of post-translational knockdown. Furthermore, we demonstrate here that nanoNEDD4-2 can selectively degrade the ion channel Q1 in situ, emphasizing the potential for customized protein manipulation with engineered E3 ligases.

One advantage of this post-translational knockdown approach is the potential to uniquely manipulate ion channel macromolecular complexes. Many ion channels, including Q1, are multi-protein complexes made up minimally of pore-forming proteins associated with accessory subunits. For example, Q1 can associate with any of five KCNE auxiliary subunits (KCNE1-KCNE5), each of which confers distinctive functional properties (*Sun et al., 2012*). Moreover, some KCNE subunits may interact with other $K^+$ channel pore-forming subunits (*Abbott et al., 1999*; *Abbott, 2016*). Cardiac ventricular myocytes express all five KCNE subunits together with Q1 (*Radicke et al., 2006*). Thus, a method to selectively inactivate protein complexes composed of specific Q1/KCNE combinations in heart cells could be powerful in illuminating the physiological logic for having multiple KCNE subunits expressed in heart cells. The capability to inactivate specific macromolecular complexes is beyond the capacity of genomic and mRNA interference approaches since simply knocking out expression of a particular KCNE could have reverberations on multiple channel types, thereby limiting specificity. By contrast, post-translational based methods such as the engineered E3 ligase approach potentially offer a platform to address this blind spot in macromolecular complex inactivation. The observation that targeting nanoCHIP to KCNE1 effectively arrests trafficking of Q1 offers rational strategies in pursuit of this goal.

Finally, it is worth commenting on potential therapeutic dimensions of our findings. Gain-of-function mutations in distinct ion channels cause diverse diseases including but not limited to: (Na$_V$1.7) inherited erythromelalgia and paroxysmal extreme pain disorder (*Waxman, 2013*); (Ca$_V$2.1) familial hemiplegic migraine (*Pietrobon, 2010*); (Ca$_V$1.2) Timothy syndrome (*Splawski et al., 2004*); (TrpV4) heritable skeletal dysplasia (*Rock et al., 2008*); (KCNQ1 and HERG) short QT syndrome and familial atrial fibrillation (*Giudicessi and Ackerman, 2012*). Targeted ubiquitination of overactive channels may provide a viable therapeutic strategy for some gain-of-function channelopathies. Although the nano-E3 ligases reported here are only effective against GFP/YFP-tagged proteins and do not target endogenous proteins, this is addressable by development of antibody mimetic proteins capable of binding specific target peptides in situ with high affinity. Various methods to develop such antibody mimetic proteins have recently emerged, including; nanobodies, single chain variable fragments (scFv), DARPins, and FingRs/monobodies (*Koide et al., 1998*; *Pardon et al., 2014*; *Plückthun, 2015*; *Gross et al., 2016*; *Sha et al., 2017*). Furthermore, a chemical strategy has been developed that utilizes hetero-bivalent small molecules referred to as PROTACS (proteolysis-targeting chimeras) to bridge endogenous substrates to endogenous ubiquitin ligases (*Schneekloth et al., 2004*; *Lai and Crews, 2017*). Our results suggest that development of PROTACS capable of selectively targeting endogenous E3 ubiquitin ligases to ion channels may be a promising therapy for diverse cardiovascular and neurological diseases.

# Materials and methods

## Key resources table

| Reagent type (species) or resource | Designation | Source or reference | Identifiers | Additional information |
|---|---|---|---|---|
| gene (human) | KCNQ1 | | NM_000218 | |
| gene (rabbit) | Cav1.2 | | NM_001136522 | |
| strain, strain background (E. coli) | XL10-Gold | Agilent | | |
| cell line (human) | HEK293 | other | RRID:CVCL_0045 | Laboratory of Robert Kass |
| cell line (human) | CHO | ATCC | RRID:CVCL_0214 | CHO-K1, ATCC, CCL-61 |
| recombinant DNA reagent | nano-P2A-CFP | this paper | | Made from GFP-nanobody (vhhGFP4) (*Kubala et al., 2010*); see molecular biology and cloning |
| recombinant DNA reagent | nanoCHIP-P2A-CFP | this paper | | Made by gene synthesis (Genewiz); see molecular biology and cloning |
| recombinant DNA reagent | nanoCHIP*-P2A-CFP | this paper | | Made by PCR; see molecular biology and cloning |
| recombinant DNA reagent | nanoNSlmb-P2A-CFP | this paper; PMID: 22157958 | | Made from pcDNA3_NSlmb-vhhGFP4 (Addgene #35579) (*Caussinus et al., 2011*); see molecular biology and cloning |
| recombinant DNA reagent | nanoMDM2-P2A-CFP | this paper | | Made from pcDNA3 MDM2 WT (Addgene #16233) (*Zhou et al., 2001*); see molecular biology and cloning |
| recombinant DNA reagent | CFP-P2A-nanoNEDD4-2 | this paper | | Made from PCI_NEDD4L (Addgene #27000) (*Gao et al., 2009*); see molecular biology and cloning |
| recombinant DNA reagent | CFP-P2A-nanoNEDD4-2* | this paper | | Made by site-directed mutagenesis of above; see molecular biology and cloning |
| recombinant DNA reagent | CFP-P2A-NEDD4-2 | this paper | | Made from PCI_NEDD4L (Addgene #27000) (*Gao et al., 2009*); see molecular biology and cloning |
| recombinant DNA reagent | iN-CHIP | this paper | | FRB:CHIP-P2A-CFP-P2A-FKBP:nano; see molecular biology and cloning |

*Continued on next page*

*Continued*

| Reagent type (species) or resource | Designation | Source or reference | Identifiers | Additional information |
|---|---|---|---|---|
| recombinant DNA reagent | BBS-KCNQ1-YFP | PMID: 25344363 | | |
| recombinant DNA reagent | BBS-KCNQ1 | this paper | | |
| recombinant DNA reagent | KCNQ1-YFP | PMID: 25344363 | | |
| recombinant DNA reagent | KCNQ1 | PMID: 25344363 | | |
| recombinant DNA reagent | KCNE1-YFP | PMID: 25344363 | | |
| recombinant DNA reagent | KCNE1 | PMID: 25344363 | | |
| recombinant DNA reagent | BBS-a1C-YFP | PMID: 20308247; PMID: 24003157 | | |
| recombinant DNA reagent | a1C-YFP | PMID: 20308247 | | |
| recombinant DNA reagent | B2a | PMID: 20308247 | | |
| biological sample (R. norvegicus) | Adult Heart Ventricular Cells | PMID: 19532115 | | |
| antibody | Anti-Q1 antibody, APC-022 | Alomone | RRID:AB_2040099 | 1:1000 |
| antibody | Anti-Ubiquitin, VU1 | LifeSensors | | 1:500 |
| peptide, recombinant protein | Protein A/G Sepharose beads | Rockland | | |
| peptide, recombinant protein | a-bungarotoxin, Alexa Fluor 647 conjugate | Life Technologies | | |
| peptide, recombinant protein | a-bungarotoxin, Biotin-XX conjugate | Life Technologies | | |
| peptide, recombinant protein | a-bungarotoxin | Life Technologies | | |
| peptide, recombinant protein | Streptavidin, Alexa Fluor 647 conjugate | Life Technologies | | |
| peptide, recombinant protein | Streptavidin, Qdot 655 conjugate | Life Technologies | | |
| commercial assay or kit | AdEasy Adenoviral Vector Systems | Stratagene | | |
| commercial assay or kit | X-tremeGENE HP DNA Transfection Reagent | Roche | | |
| commercial assay or kit | QuikChange Lightning Site-Directed Mutagenesis Kit | Stratagene | | |
| chemical compound, drug | Rapamycin | Sigma | | |
| software, algorithm | FlowJo | | RRID:SCR_008520 | |
| software, algorithm | PulseFit | HEKA | | |
| software, algorithm | Origin | | RRID:SCR_014212 | |
| software, algorithm | Graphpad Prism | | RRID:SCR_002798 | |

## Molecular biology and cloning of plasmid vectors

A customized bicistronic vector (xx-P2A-CFP) was synthesized in the pUC57 vector, in which coding sequence for P2A peptide was sandwiched between an upstream multiple cloning site and enhanced cyan fluorescent protein (CFP) (Genewiz, South Plainfield, NJ). The xx-P2A-CFP fragment was amplified by PCR and cloned into the PiggyBac CMV mammalian expression vector (System Biosciences, Palo Alto, CA) using NheI/NotI sites. To generate nano-xx-P2A-CFP, we PCR amplified the coding sequence for GFP nanobody (vhhGFP4) and cloned it into xx-P2A-CFP using NheI/AflII sites. The nanoCHIP construct was created by gene synthesis (Genewiz), and featured the coding sequence for GFP nanobody (vhhGFP4) (*Kubala et al., 2010*) in frame with the minimal catalytic unit of CHIP E3 ligase (residues 128–303), separated by a flexible GSG linker. This fragment was amplified by PCR and cloned into the xx-P2A-CFP vector using NheI/AflII sites. To create the catalytically inactive nanoCHIP*, we deleted the coiled-coil domain [Δ128–229] as previously described (*Nikolay et al.,*

2004), by amplifying the U-box domain of CHIP E3 ligase (residues 230–303), and cloned this fragment into nano-xx-P2A-CFP using AscI/AflII sites separated by a flexible GSG linker.

NSlmb:nano-P2A-CFP was derived from pcDNA3_NSlmb-vhhGFP4 (Addgene #35579, Cambridge, MA) (Caussinus et al., 2011). We PCR amplified the NSlmb-vhhGFP4 fragment and cloned it into xx-P2A-CFP using NheI/AflII sites. To generate nanoMDM2, we PCR amplified the RING domain (residues 432–491) from MDM2 (Addgene #16233) (Zhou et al., 2001) and cloned this fragment into nano-xx-P2A-CFP using AscI/AflII sites. To create nanoNEDD4L we first PCR amplified the HECT domain (residues 596–975) of NEDD4L (PCI_NEDD4L; Addgene #27000) (Gao et al., 2009) and cloned this fragment into nano-xx-P2A-CFP using AscI/AflII sites. The resulting construct, nanoNEDD4L-P2A-CFP expressed poorly so we swapped positions of the nanoNEDD4L and CFP. We first generated CFP-P2A-xx and then PCR amplified nanoNEDD4L. The resulting fragment was cloned into CFP-P2A-xx using BglII/NotI sites. To create the catalytically inactive nanoNEDD4-2*, we introduced a point mutation at the catalytic cysteine residue [C942S] by site-directed mutagenesis.

KCNQ1/E1 constructs were made as described previously (Aromolaran et al., 2014). Briefly, overlap extension PCR was used to fuse enhanced yellow fluorescent proteins (EYFP) in frame to the C-terminus of KCNQ1 and KCNE1. A 13-residue bungarotoxin-binding site (BBS; TGGCGGTACTAC-GAGAGCAGCCTGGAGCCCTACCCCGAC) (Sekine-Aizawa and Huganir, 2004; Yang et al., 2010) was introduced between residues 148–149 in the extracellular S1–S2 loop of KCNQ1 using the Quik-Change Lightning Site-Directed Mutagenesis Kit (Stratagene, La Jolla, CA) according to the manufacturer's instructions.

The inducible nanoCHIP construct (FRB:CHIP-P2A-CFP-P2A-FKBP:nano) was created in three parts. First, FRB:CHIP-P2A-CFP was created by PCR amplifying the CHIP catalytic domain and cloning the amplified fragment into FRBxx-P2A-CFP vector using AscI/AflII sites. Second, we used overlap extension PCR to create a P2A-FKBP:nano cassette which was then cloned downstream of CFP in the FRB:CHIP-P2A-CFP construct using BglII/NotI sites, generating FRB:CHIP-P2A-CFP-P2A-FKBP-nano.

## Generation of adenoviral vectors

Adenoviral vectors were generated using the pAdEasy system (Stratagene) according to manufacturer's instructions as previously described (Subramanyam et al., 2013; Aromolaran et al., 2014). Plasmid shuttle vectors (pShuttle CMV) containing cDNA for nano-P2A-CFP, nanoCHIP-P2A-CFP, and BBS-Q1-YFP were linearized with PmeI and electroporated into BJ5183-AD-1 electrocompetent cells pre-transformed with the pAdEasy-1 viral plasmid (Stratagene). PacI restriction digestion was used to identify transformants with successful recombination. Positive recombinants were amplified using XL-10-Gold bacteria, and the recombinant adenoviral plasmid DNA linearized with PacI digestion. HEK cells cultured in 60 mm diameter dishes at 70–80% confluency were transfected with PacI-digested linearized adenoviral DNA. Transfected plates were monitored for cytopathic effects (CPEs) and adenoviral plaques. Cells were harvested and subjected to three consecutive freeze-thaw cycles, followed by centrifugation (2,500 × g) to remove cellular debris. The supernatant (2 mL) was used to infect a 10 cm dish of 90% confluent HEK293 cells. Following observation of CPEs after 2–3 d, cell supernatants were used to re-infect a new plate of HEK293 cells. Viral expansion and purification was carried out as previously described (Colecraft et al., 2002). Briefly, confluent HEK293 cells grown on 15 cm culture dishes (x8) were infected with viral supernatant (1 mL) obtained as described above. After 48 hr, cells from all of the plates were harvested, pelleted by centrifugation, and resuspended in 8 mL of buffer containing (in mM) Tris·HCl 20, CaCl$_2$ 1, and MgCl$_2$ 1 (pH 8.0). Cells were lysed by four consecutive freeze-thaw cycles and cellular debris pelleted by centrifugation. The virus-laden supernatant was purified on a cesium chloride (CsCl) discontinuous gradient by layering three densities of CsCl (1.25, 1.33, and 1.45 g/mL). After centrifugation (50,000 rpm; SW41Ti Rotor, Beckman-Coulter Optima L-100K ultracentrifuge; 1 hr, 4°C), a band of virus at the interface between the 1.33 and 1.45 g/mL layers was removed and dialyzed against PBS (12 hr, 4°C). Adenoviral vector aliquots were frozen in 10% glycerol at −80°C until use.

## Cell culture and transfections

Human embryonic kidney (HEK293) cells were a kind gift from the laboratory of Dr. Robert Kass (Columbia University). Cells were mycoplasma free, as determined by the MycoFluor Mycoplasma

Detection Kit (Invitrogen, Carlsbad, CA). Low passage HEK293 cells were cultured at 37°C in DMEM supplemented with 8% fetal bovine serum (FBS) and 100 mg/mL of penicillin–streptomycin. HEK293 cell transfection was accomplished using the calcium phosphate precipitation method. Briefly, plasmid DNA was mixed with 62 μL of 2.5M $CaCl_2$ and sterile deionized water (to a final volume of 500 μL). The mixture was added dropwise, with constant tapping to 500 μL of 2x Hepes buffered saline containing (in mM): Hepes 50, NaCl 280, $Na_2HPO_4$ 1.5, pH 7.09. The resulting DNA–calcium phosphate mixture was incubated for 20 min at room temperature and then added dropwise to HEK293 cells (60–80% confluent). Cells were washed with $Ca^{2+}$-free phosphate buffered saline after 4–6 hr and maintained in supplemented DMEM.

Chinese hamster ovary (CHO) cells were obtained from ATCC (Manassas, VA), and cultured at 37°C in Kaighn's Modified Ham's F-12K (ATCC) supplemented with 8% FBS and 100 mg/mL of penicillin–streptomycin. CHO cells were transiently transfected with desired constructs in 35 mm tissue culture dishes—KCNQ1 (0.5 μg), KCNE1 (0.5 μg), and nano-P2A-CFP (0.5 μg), and nanoCHIP-P2A-CFP (0.5 μg) using X-tremeGENE HP (1:2 DNA/reagent ratio) according to the manufacturers' instructions (Roche, Indianapolis, IN).

Primary cultures of adult rat heart ventricular cells were prepared as previously described (*Colecraft et al., 2002*; *Subramanyam et al., 2013*), in accordance with the guidelines of Columbia University Animal Care and Use Committee. Adult male Sprague–Dawley rats were euthanized with an overdose of isoflurane. Hearts were excised and ventricular myocytes isolated by enzymatic digestion with 1.7 mg Liberase–TM enzyme mix (Roche) using a Langendorff perfusion apparatus. Healthy rod-shaped myocytes were cultured in Medium 199 (Life Technologies) supplemented with (in mM) carnitine (5), creatine (5), taurine (5) penicillin-streptomycin-glutamine (0.5%, Life technologies), and 5% (vol/vol) FBS (Life Technologies) to promote attachment to dishes. After 5 hr, the culture medium was switched to Medium 199 with 1% (vol/vol) serum, but otherwise supplemented as described above. Cultures were maintained in humidified incubators at 37°C and 5% $CO_2$.

## Flow cytometry assay of total and surface Q1 channels

Cell surface and total ion channel pools were assayed by flow cytometry in live, transfected HEK293 cells as previously described (*Yang et al., 2010*; *Aromolaran et al., 2014*). Briefly, 48 hr post-transfection, cells cultured in 6-well plates gently washed with ice cold PBS containing $Ca^{2+}$ and $Mg^{2+}$ (in mM: 0.9 $CaCl_2$, 0.49 $MgCl_2$, pH 7.4), and then incubated for 30 min in blocking medium (DMEM with 3% BSA) at 4°C. HEK293 cells were then incubated with 1 μM Alexa Fluor 647 conjugated α-bungarotoxin ($BTX_{647}$; Life Technologies) in DMEM/3% BSA on a rocker at 4°C for 1 hr, followed by washing three times with PBS (containing $Ca^{2+}$ and $Mg^{2+}$). Cells were gently harvested in $Ca^{2+}$-free PBS, and assayed by flow cytometry using a BD LSRII Cell Analyzer (BD Biosciences, San Jose, CA, USA). CFP- and YFP-tagged proteins were excited at 407 and 488 nm, respectively, and Alexa Fluor 647 was excited at 633 nm.

Optical pulse chase assays to monitor rates of channel forward trafficking and internalization were conducted on live, transfected HEK293 cells. For the iN-CHIP treatment groups, cells were pretreated with 1 μM rapamycin for 3 hr prior to the experiments. Cells were placed on 4°C to halt trafficking processes and washed twice with PBS containing $Ca^{2+}$ and $Mg^{2+}$. For forward trafficking experiments, cells were incubated with 5 μM untagged BTX in DMEM/3% BSA at 4°C for 1 hr to block surface channels, and then washed three times with PBS containing $Ca^{2+}$ and $Mg^{2+}$. Cells were incubated with DMEM/3% BSA and placed at 37°C to resume trafficking for different time intervals (0, 5, 10, 20, 40, 60 min). Cells were then returned to 4°C and newly delivered channels were labeled with 1 μM $BTX_{647}$ in DMEM/3% BSA for 1 hr. Finally, cells were washed three times with PBS containing $Ca^{2+}$ and $Mg^{2+}$, gently harvested in $Ca^{2+}$-free PBS, and assayed by flow cytometry. For internalization experiments, cells were incubated in DMEM/3% BSA blocking medium for 30 min at 4°C, followed by a pulse of 1 μM biotinylated α-bungarotoxin (BTX-biotin; Life Technologies) for 1 hr with gentle rocking at 4°C. Cells were washed three times in PBS containing $Ca^{2+}$ and $Mg^{2+}$ and placed in DMEM/3% BSA at 37°C for different time intervals (0, 5, 10, 20, 40, 60 min) to resume trafficking. Cells were returned to 4°C, washed once with PBS, and channels remaining at the surface were labeled with streptavidin-conjugated Alexa Fluor 647 (Life Technologies). Finally, cells were washed twice more with PBS with $Ca^{2+}$ and $Mg^{2+}$, harvested in $Ca^{2+}$-free PBS, and assayed by flow cytometry.

## Confocal detection of total and surface Q1 expression in cardiomyocytes

At 48 hr post-infection, adult rat cardiomyocytes cultured on 35 mm MatTek dishes (MatTek Corporation, Ashland, MA) were gently washed with M199 media (with 0.9 mM $CaCl_2$, 0.49 mM $MgCl_2$, pH 7.4) and fixed with 4% formaldehyde for 10 min at room temperature (RT). Cardiomyocytes were washed three times with PBS, and incubated for 30 min in blocking medium (M199 with 3% BSA). Cardiomyocytes were then incubated with 1 µM BTX-biotin in M199/3% BSA at room temperature for 1 hr followed by washing three times with PBS to remove unbound biotinylated BTX. Cells were then incubated with 10 nM streptavidin-conjugated Quantum Dot 655 ($QD_{655}$; Life Technologies) for 1 hr at 4°C in the dark, washed three times with PBS, and imaged with Nikon Ti Eclipse inverted microscope for scanning confocal microscopy.

## Electrophysiology

For potassium channel measurements, whole-cell membrane currents were recorded at room temperature in CHO cells using an EPC-10 patch-clamp amplifier (HEKA Electronics, Lambrecht/Pfalz, Germany) controlled by the PatchMaster software (HEKA). A coverslip with adherent CHO cells was placed on the glass bottom of a recording chamber (0.7–1 mL in volume) mounted on the stage of an inverted Nikon Eclipse Ti-U microscope. Micropipettes were fashioned from 1.5 mm thin-walled glass and fire-polished. Internal solution contained (mM): 133 KCl, 0.4 GTP, 10 EGTA, 1 $MgSO_4$, 5 $K_2ATP$, 0.5 $CaCl_2$, and 10 HEPES (pH 7.2). External solution contained (in mM): 147 NaCl, 4 KCl, 2 $CaCl_2$, and 10 HEPES (pH 7.4). Pipette resistance was typically 1.5 MΩ when filled with internal solution. I–V curves were generated from a family of step depolarizations (−40 to +100 mV in 10 mV steps from a holding potential of −50 mV). Currents were sampled at 20 kHz and filtered at 5 kHz. Traces were acquired at a repetition interval of 10 s.

For calcium channel measurements, whole-cell recordings were carried out in HEK293 cells at room temperature. Internal solution contained (mM): 135 Cs Methanesulfonate, 5 CsCl, 5 EGTA, 1 $MgCl_2$, 4 MgATP, 10 HEPES (pH 7.2). External solution contained (mM): 140 tetraethylammonium-methanesulfonate, 5 $BaCl_2$, 10 HEPES (pH 7.4). Leak and capacitive currents were subtracted using a P/4 protocol. I-V curves were generated from a family of step depolarizations (−60 to +100 mV in 10 mV steps from a holding potential of −90 mV). Currents were sampled at 20 kHz and filtered at 5 kHz. Traces were acquired at a repetition interval of 10 s.

## Immunoprecipitation and Western blotting

HEK293 cells were washed once with PBS without $Ca^{2+}$, harvested, and resuspended in RIPA lysis buffer containing (in mM) Tris (20, pH 7.4), EDTA (1), NaCl (150), 0.1% (wt/vol) SDS, 1% Triton X-100, 1% sodium deoxycholate and supplemented with protease inhibitor mixture (10 µL/ mL, Sigma-Aldrich, St. Louis, MO), PMSF (1 mM, Sigma-Aldrich), and PR-619 deubiquitinase inhibitor (50 µM, LifeSensors, Malvern, PA). Lysates were prepared by incubation at 4°C for 1 hr, with occasional vortex, and cleared by centrifugation (10,000 × g, 10 min, 4°C). Supernatants were transferred to new tubes, with aliquots removed for quantification of total protein concentration determined by the bis-cinchonic acid protein estimation kit (Pierce Technologies, Waltham, MA). Lysates were pre-cleared by incubation with 10 µL Protein A/G Sepharose beads (Rockland) for 40 min at 4°C and then incubated with 0.75 µg anti-Q1 (Alomone, Jerusalem, Israel) for 1 hr at 4°C. Equivalent total protein amounts were added to spin-columns containing 25 µL Protein A/G Sepharose beads, tumbling overnight at 4°C. Immunoprecipitates were washed 3–5 times with RIPA buffer, spun down at 500 × g, eluted with 40 µL of warmed sample buffer [50 mM Tris, 10% (vol/vol) glycerol, 2% SDS, 100 mM DTT, and 0.2 mg/mL bromophenol blue], and boiled (55°C, 15 min). Proteins were resolved on a 4–12% Bis·Tris gradient precast gel (Life Technologies) in Mops-SDS running buffer (Life Technologies) at 200 V constant for ~1 hr. We loaded 10 µL of the PageRuler Plus Prestained Protein Ladder (10–250 kDa, Thermo Fisher, Waltham, MA) alongside the samples. Protein bands were transferred by tank transfer onto a nitrocellulose membrane (3.5 hr, 4°C, 30 V constant) in transfer buffer (25 mM Tris pH 8.3, 192 mM glycine, 15% (vol/vol) methanol, and 0.1% SDS). The membranes were blocked with a solution of 5% nonfat milk (BioRad) in tris-buffered saline-tween (TBS-T) (25 mM Tris pH 7.4, 150 mM NaCl, and 0.1% Tween-20) for 1 hr at RT and then incubated overnight at 4°C with primary antibodies (anti-Q1, Alomone) in blocking solution. The blots were washed with TBS-T

three times for 10 min each and then incubated with secondary horseradish peroxidase-conjugated antibody for 1 hr at RT. After washing in TBS-T, the blots were developed with a chemiluminiscent detection kit (Pierce Technologies) and then visualized on a gel imager. Membranes were then stripped with harsh stripping buffer (2% SDS, 62 mM Tris pH 6.8, 0.8% ß-mercaptoethanol) at 50°C for 30 min, rinsed under running water for 2 min, and washed with TBST (3x, 10 min). Membranes were pre-treated with 0.5% glutaraldehyde and re-blotted with anti-ubiquitin (VU1, LifeSensors) as per the manufacturers' instructions.

### Data and statistical analyses

Data were analyzed off-line using FlowJo, PulseFit (HEKA), Microsoft Excel, Origin and GraphPad Prism software. Statistical analyses were performed in Origin or GraphPad Prism using built-in functions. Statistically significant differences between means ($p < 0.05$) were determined using Student's $t$ test for comparisons between two groups. Data are presented as means $\pm$ s.e.m.

## Acknowledgements

We thank Drs. Gerhard Thiel (Technische Universität Darmstadt) and Anna Moroni (University of Milan) for comments on the manuscript; Ming Chen and Dr. Papiya Choudhury for excellent technical support. This work was supported by grants RO1-HL121253 and 1RO1-HL122421 from the NIH (to HMC). SK was supported by a Medical Scientist Training Program grant (T32 GM007367). Flow cytometry experiments were performed in the CCTI Flow Cytometry Core, supported in part by the NIH (S10RR027050). Confocal images were collected in the HICCC Confocal and Specialized Microscopy Shared Resource, supported by NIH (P30 CA013696).

## Additional information

### Funding

| Funder | Grant reference number | Author |
| --- | --- | --- |
| National Heart, Lung, and Blood Institute | RO1-HL121253 | Henry M Colecraft |
| National Heart, Lung, and Blood Institute | 1RO1-HL122421 | Henry M Colecraft |
| National Institutes of Health | T32 GM007367 | Scott A Kanner |

The funders had no role in study design, data collection and interpretation, or the decision to submit the work for publication.

### Author contributions

Scott A Kanner, Conceptualization, Data curation, Formal analysis, Investigation, Writing—original draft, Writing—review and editing; Travis Morgenstern, Data curation, Formal analysis, Writing—review and editing; Henry M Colecraft, Conceptualization, Data curation, Formal analysis, Supervision, Funding acquisition, Investigation, Writing—original draft, Writing—review and editing

### Author ORCIDs

Scott A Kanner http://orcid.org/0000-0002-1579-8858
Travis Morgenstern http://orcid.org/0000-0003-2634-8470
Henry M Colecraft http://orcid.org/0000-0002-2340-8899

### Ethics

Animal experimentation: Primary cultures of adult rat heart ventricular cells were prepared as previously described (Colecraft et al., 2002; Subramanyam et al., 2013), in accordance with the guidelines of Columbia University Animal Care and Use Committee. All of the animals were handled according to approved institutional animal care and use committee (IACUC) protocols (# AC-AAAS2515).

Decision letter and Author response
Decision letter https://doi.org/10.7554/eLife.29744.022
Author response https://doi.org/10.7554/eLife.29744.023

## Additional files

**Supplementary files**
• Transparent reporting form
DOI: https://doi.org/10.7554/eLife.29744.020

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
