## [Decision Letter]

[Editors’ note: this article was originally rejected after discussions between the reviewers, but the authors were invited to resubmit after an appeal against the decision.]

Thank you for submitting your work entitled "Sculpting ion channel functional expression with engineered ubiquitin ligases" for consideration by *eLife*. Your article has been reviewed by three peer reviewers, and the evaluation has been overseen by a Reviewing Editor and a Senior Editor. Two reviewers have agreed to reveal their identity: Hugues Abriel (Reviewer 1) and Jim Trimmer (Reviewer 3).

Our decision has been reached after consultation between the reviewers. Based on these discussions and the individual reviews below, we regret to inform you that your work will not be considered further for publication in *eLife*.

Your study describes a clever system to regulate membrane protein expression at the cell surface and also provides a degree of temporal control that will be useful to study the role of these proteins in various cellular signaling pathways. The primary common concern of the reviewers is that the manuscript in its present form is mainly methodological without a biologically significant insight. In our opinion, it is unrealistic to expect that any new data to address these concerns can be added within the timeframe of a normal revision process. Please do not hesitate to contact us if you have questions.

*Reviewer #1:*

This manuscript by Kanner and colleagues presents results of sophisticated experiments where the authors used "engineered" proteins to study the consequences of ubiquitination of ion channels, in particular, the KCNQ1 potassium channel by the ubiquitin ligase CHIP. This is done by adding tags and protein-protein interaction motives to these proteins and looking at either their total cellular expression or specific membrane expression. The paper not only describes the CHIP-dependent ubiquitination of KCNQ1, but also its ubiquitination by other E3 ligases such as the NEDD4-2 ligase, already shown to ubiquitinate KCNQ1. This cellular system was also active on a more "complex" ion channel, i.e. the Cav1.2 large protein.

The presented data are in general well presented and convincing. Some controls are missing, however, such as dead-E3 ligase or mutation of the PY motif of KCNQ1.

Overall the system developed by the authors is convincing and potentially interesting to elucidate the specific and distinct roles of E3 ligases targeting ion channels.

Substantive concerns:

The main serious concern related to this work is that at this stage this is "only" a methodological paper describing a sophisticated way to artificially ubiquitinate a target, in this case the KCNQ1 channel. What do we learn? The only original new finding would be that the reduced membrane expression of KCNQ1 when CHIP is active and co-expressed seems to be caused by reduced forward trafficking. Note that this effect could be challenged with the use of the brefeldin A.

I think that this approach may have been useful to address many open questions as to the "site of action" of the different ligases, the types of ubiquitination of the specific ligases, and others. Another problem is the fact that all these systems are still very "artificial" and all what will be learned can be very different from really native conditions.

*Reviewer #2:*

In this manuscript, Kanner et al. fused the catalytic domain of the E3 ligase CHIP to nanobody recognizing GFP/YFP and targeted this nanoCHIP to YFP-tagged KCNQ1 +/- its associated subunit KCNE1 (a K^+^ channel defective in LongQT) to ubiquitinate and degrade or endocytose this channel in cells, in order to generate a tool to regulate/eliminate this channel at will. They found that this maneuver reduced cell surface levels of this K^+^ channel by decreasing its forward trafficking to the plasma membrane, but not increasing its internalization.

Evaluation:

The authors developed a neat system to post-translationally regulate PM protein levels at will, and moreover, generated an inducible system that enables temporal control of targeting of these TM proteins. This is a very useful tool that can be of great benefit for future studies. This paper, however, has some issues, as follows:

1) Novelty: There have been quite a few papers published (several of them cited by the authors) that already describe various systems to regulate cellular stability of proteins at will by targeting them for degradation by E3 ligases (including CHIP). Some of them actually tested the system in vivo (in mice). The paper here combines some of the published methods into an improved one, but the idea is not novel, nor use of CHIP to achieve the goal.

2) The proteins targeted are not necessarily the ones that are the cognate substrates for the chosen E3 ligase. While this is OK to get rid of a specific protein (akin to knockdown or knockout, but with better regulation post-translationally) for analyses such as cell fate, survival/proliferation, or other whole cell function(s), using this method for analysis of regulation of trafficking and endocytosis of a channel that is artificially targeted in this manner in cells may be problematic. In fact, it is probably not a surprise that CHIP regulates forward trafficking of this channel (based on its known function in cells), but not its endocytosis. Endogenously, it is quite likely that other E3 ligases (e.g. Nedd4-2) regulate Q1 endocytosis. Indeed, Figure 3—figure supplement 1 actually suggests that i.e., the biological significance of studying trafficking/endocytosis of a PM protein with an artificial E3 ligase (that anyway only weakly ubiquitinates it) is not clear.

3) Adding a tag to proteins (YFP here) requires their overexpression, thus creating an artifact. The authors suggest that the method they created can be used with antibodies to endogenous proteins, but (i) they have not done it, and (ii) such a system will rely on exquisitely specific antibodies (or parts of them), which are not easy to generate, let alone prove that they target a single protein in a cell.

4) CHIP-mediated ubiquitination of Q1 (Figure 3) is very weak and not convincing (and the experiments lack loading controls).

5) The data in Figure 3—figure supplement 1 (showing Nedd4-2 mediated enhanced removal of Q1 from the cell surface) are not in agreement with those shown in Figure 2 and described in the subsection “nanoCHIP abolishes Q1 surface population, with modest effect on total channel pool”. Please explain the discrepancy.

6) Why was channel function analysis (Figure 5) done on CHO cells while all biochemical/IF studies on 293 cells?

*Reviewer #3:*

The manuscript by Kanner and Colecraft focuses on the development of a novel and innovative technique for modulating expression of membrane proteins. They utilize a fusion between an intrabody (in this case a nanobody) with the catalytic domain of an E3 ubiquitin ligase to generate a "nanochip" capable of specifically modulating target protein activity. They show specific downregulation of activity of their target of choice, the KCNQ1 ion channel, by targeting wither the principle KCNQ1 or auxiliary KCNE1 subunit. They provide data to support a model whereby the downregulation occurs via loss of cell surface expression of the ion channel (that is, a decrease in n), as opposed to downregulating the activity of cell surface channels (via effects on open probability or single channel conductance). They also provide support for a mechanism whereby it is cell surface delivery of the ion channels that is the basis of the nanochip-mediated downregulation. The data are generally of high quality (although see concerns below), and are not overinterpreted. The writing is very clear, and the figures are for the most part of high quality and clear.

One concern with the data is that in the flow cytometry plots (Figure 2, Figure 4, Figure 6 and Figure 9), there are substantial populations of KCNQ1expressing cells with YFP but not CFP signals, and vice versa. There are many cells with levels of CFP at the same level of those with maximal YFP levels, but that have no YFP (those in the upper left quadrant), and vice-versa (in the lower right quadrant). This also holds for BTX labeling, with lots of cells with one or the other. This does not make sense given how the authors describe the experiments. In both cases I would have expected something more like a diagonal line, with the origin in the lower left, and then going towards the upper right, such that the levels of the two labels are correlated. This also holds for the analysis of the KCNE1 expression and effects of the nanochips (Figure 4).

One other concern is that it appears that nanochip expression lowers the levels of CFP in expressing cells (and the representation of CFP expressing cells in the population). This should not be the case and suggests non-specific effects of the nanochips on protein expression.

Given these apparent discrepancies greatly strengthen the paper. This would confirm loss of cell surface expression of KCNQ1 by a technique independent of flow cytometry, which is used exclusively to assay levels of cell surface expression (that is, n) independent of effects of nanochips on channel open probability or single channel conductance. Biotinylation analyses could be combined with the type of anti-ubiquitin blots already presented to further support that ubiquitinated proteins are internalized.

---

## [Author Response]

[Editors’ note: the author responses to the first round of peer review follow.]

Reviewer #1:[…] The presented data are in general well presented and convincing. Some controls are missing, however, such as dead-E3 ligase or mutation of the PY motif of KCNQ1.

We have included new data with a deletion mutant of CHIP that has been previously demonstrated to ablate its E3 ligase activity (Figure 2—figure supplement 2. We also have now included new control data with a catalytically inactive nanoNEDD4-2 mutant (Figure 2—figure supplement 4). In the text, these new data are described in the subsection “nanoCHIP abolishes Q1 surface population, with modest effect on total channel pool”.

Mutating the PY motif is not really applicable to the nanoE3 ligases because they do not utilize this motif to target the channel. Rather, they target YFP fused to the channel. Hence, the appropriate controls are channels in which YFP is omitted, and these are included in the manuscript (Figure 2, Figure 4 and Figure 5).

Overall the system developed by the authors is convincing and potentially interesting to elucidate the specific and distinct roles of E3 ligases targeting ion channels.Substantive concerns:The main serious concern related to this work is that at this stage this is "only" a methodological paper describing a sophisticated way to artificially ubiquitinate a target, in this case the KCNQ1 channel.

The editors of *eLife* have indicated the manuscript will be considered under the Tools and Resources category, for which a methodological paper is appropriate. We concur with this decision.

We do believe that the method we have introduced has great potential to revolutionize studies of ubiquitin regulation of ion channels in situ, given the complexities of the ubiquitin code and ion channels themselves. That the reviewer writes “I think that this approach may have been useful to address many open questions as to the site of action of the different ligases, the types of ubiquitination of the specific ligases, and others” is actually proof of the point of the enabling nature of the work. Indeed, we are pursuing many of the questions that the reviewer has posed, which are not trivial, and are beyond the scope of the present study.

What do we learn? The only original new finding would be that the reduced membrane expression of KCNQ1 when CHIP is active and co-expressed seems to be caused by reduced forward trafficking. Note that this effect could be challenged with the use of the brefeldin A.I think that this approach may have been useful to address many open questions as to the "site of action" of the different ligases, the types of ubiquitination of the specific ligases, and others. Another problem is the fact that all these systems are still very "artificial" and all what will be learned can be very different from really native conditions.

We have sharpened our Discussion to emphasize what we believe are several original findings of the work. An example is the following paragraph:

“Without exception, all previous renditions of this technology have relied on degradation of target proteins as the ultimate expression of efficacy. […] Furthermore, we demonstrate here that nanoNEDD4-2 can selectively degrade the ion channel Q1 in situ, emphasizing the potential for customized protein manipulation with engineered E3 ligases.”

In addition, we believe there are other non-trivial new results from the paper. For example, the finding that we can reduce surface density of the channel complex by targeting an E3 ligase to the auxiliary subunit is new and has major implications for designing approaches to manipulate ion channel macromolecular complexes that are presently non-existent (subsection “Engineered E3 ligase approach as a tool to manipulate functional expression of membrane proteins”, fourth paragraph).

Reviewer #2:[…] Evaluation:The authors developed a neat system to post-translationally regulate PM protein levels at will, and moreover, generated an inducible system that enables temporal control of targeting of these TM proteins. This is a very useful tool that can be of great benefit for future studies. This paper, however, has some issues, as follows:1) Novelty: There have been quite a few papers published (several of them cited by the authors) that already describe various systems to regulate cellular stability of proteins at will by targeting them for degradation by E3 ligases (including CHIP). Some of them actually tested the system in vivo (in mice). The paper here combines some of the published methods into an improved one, but the idea is not novel, nor use of CHIP to achieve the goal.

As the reviewer notes, we do not claim to be the first ones to target catalytic domains of E3 ligases to substrates. However, we do claim primacy in the application of this approach to ion channels, a specialized class of proteins that rely on a very different post-translational lifecycle of maturation, sorting, and trafficking. As the reviewer also notes, *all* the previous renditions of this technology has relied on degradation of target proteins as the ultimate expression of efficacy. That we can achieve effective functional knockdown of ion channels without degradation is a fundamental, novel and non-trivial distinction from previous studies. The combination of acute temporal control over the targeted ubiquitination process and optical pulse chase approaches to assess relative impact on distinct trafficking processes is also new.

2) The proteins targeted are not necessarily the ones that are the cognate substrates for the chosen E3 ligase. While this is OK to get rid of a specific protein (akin to knockdown or knockout, but with better regulation post-translationally) for analyses such as cell fate, survival/proliferation, or other whole cell function(s), using this method for analysis of regulation of trafficking and endocytosis of a channel that is artificially targeted in this manner in cells may be problematic. In fact, it is probably not a surprise that CHIP regulates forward trafficking of this channel (based on its known function in cells), but not its endocytosis. Endogenously, it is quite likely that other E3 ligases (e.g. Nedd4-2) regulate Q1 endocytosis. Indeed, Figure 3—figure supplement 1 actually suggests that i.e., the biological significance of studying trafficking/endocytosis of a PM protein with an artificial E3 ligase (that anyway only weakly ubiquitinates it) is not clear.

Our argument in the paper is that promiscuity among E3 ligase/substrate interactions makes it difficult to define the full complement of E3 ligases that are capable of acting on a particular substrate, and also the complement of proteins that a particular E3 ligase acts upon. The method is a useful tool for simplifying some of these complexities so that specific questions about how ubiquitination regulates diverse aspects of ion channel fate and function can be asked and answered. Moreover, the method offers an approach to study the putative in situ functions of distinct E3 ligases with known enzymatic characteristics.

The fact that nanoCHIP only weakly ubiquitinates the channel is not necessarily a weakness. That this relatively modest ubiquitination can have such flagrant effects on protein sub-cellular localization (>90% decrease in surface density) is the new insight that this particular result provides. It clearly goes against the assumption apparently made by the reviewer (and ourselves coming into the work) that more ubiquitin correlates to larger effects.

3) Adding a tag to proteins (YFP here) requires their overexpression, thus creating an artifact. The authors suggest that the method they created can be used with antibodies to endogenous proteins, but (i) they have not done it, and (ii) such a system will rely on exquisitely specific antibodies (or parts of them), which are not easy to generate, let alone prove that they target a single protein in a cell.

The reviewer is correct that the studies utilize heterologous expression of tagged ion channels which does have drawbacks, but is, nevertheless, a useful system for proof-of-principle studies such as those conducted in the manuscript. Though we have not generated intrabodies to membrane proteins (we are in the middle of this and have in fact recently made some good progress with nanobodies to cytosolic proteins for a different project) others have, including nanobodies and monobodies. That we have not done this ourselves so far is not germane to whether the approach can be generally useful to the field.

4) CHIP-mediated ubiquitination of Q1 (Figure 3) is very weak and not convincing (and the experiments lack loading controls).

Please see point #2 above with respect to this actually being a strength rather than a weakness. These are pulldown experiments for which a loading control per se does not make sense. Rather, we normalize the ubiquitin staining to the intensity of the pulled down channel. This normalization represents the most appropriate control for this type of experiment.

5) The data in Figure 3—figure supplement 1 (showing Nedd4-2 mediated enhanced removal of Q1 from the cell surface) are not in agreement with those shown in Figure 2 and described in the subsection “nanoCHIP abolishes Q1 surface population, with modest effect on total channel pool”. Please explain the discrepancy.

There is no discrepancy here. Figure 2 shows the data with nanoCHIP, which we contrast with wild-type Nedd4-2 which does decrease both surface density and total KCNQ1 expression. Subsection “nanoCHIP abolishes Q1 surface population, with modest effect on total channel pool” refers to nanoNedd4-2 (now Figure 2—figure supplement 4) which behaves similar to wt Nedd4-2 (and different from nanoCHIP). We suggest the differences between nanoCHIP and nanoNedd4-2 illustrates a strength of the approach as it allows evaluation of distinctive functional effects of different E3 ligases on an ion channel.

To clarify this point, we have sharpened our discussion of the differences between nanoCHIP and nanoNEDD4-2 with respect to their relative impact on Q1 stability in the third paragraph of the subsection “Complexities in decoding ubiquitin regulation of membrane proteins”. This discussion culminates in the following conclusions:

“Overall, these results emphasize that distinct ligases can differentially impact the stability and subcellular localization of ion channels. Moreover, this work illustrates how engineered E3 ligases can be utilized to systematically and selectively probe the impact of particular E3 ligases on target proteins in the complex cellular environment.”

Also relevant to this point:

“Without exception, all previous renditions of this technology have relied on degradation of target proteins as the ultimate expression of efficacy. […] Furthermore, we demonstrate here that nanoNEDD4-2 can selectively degrade the ion channel Q1 *in situ*, emphasizing the potential for customized protein manipulation with engineered E3 ligases.”

6) Why was channel function analysis (Figure 5) done on CHO cells while all biochemical/IF studies on 293 cells?

We have traditionally conducted electrophysiological studies of KCNQ1 channels in CHO cells, because some HEK293 cells can have a small level of endogenous KCNQ1 channels. This would not affect the flow cytometry experiments which detect tagged exogenously-expressed channels. To address the reviewer’s concern we performed electrophysiological experiments in HEK293 cells expressing Q1-YFP + KCNE1 with either nanobody alone, or nanoCHIP (Figure 5—figure supplement 1). The results showed that nanoCHIP was similarly effective in reducing *I*_Ks_ in HEK293 cells as it was in CHO cells.

Reviewer #3:[…] One concern with the data is that in the flow cytometry plots (Figure 2, Figure 4, Figure 6 and Figure 9), there are substantial populations of KCNQ1expressing cells with YFP but not CFP signals, and vice versa. There are many cells with levels of CFP at the same level of those with maximal YFP levels, but that have no YFP (those in the upper left quadrant), and vice-versa (in the lower right quadrant). This also holds for BTX labeling, with lots of cells with one or the other. This does not make sense given how the authors describe the experiments. In both cases I would have expected something more like a diagonal line, with the origin in the lower left, and then going towards the upper right, such that the levels of the two labels are correlated. This also holds for the analysis of the KCNE1 expression and effects of the nanochips (Figure 4).

We appreciate the reviewer’s questions about the flow cytometry method. We interpret the cases of cell populations having either CFP or YFP/BTX fluorescent signals, but not both, as reflecting the stochastic nature of transient transfections (and protein expression) with multiple plasmids (i.e. some cells may have one or other of the plasmid, but not both). Actually, we view this as a strength of the approach as we can see and differentiate our analysis among the different populations of cells in a way that is not accessible by conventional biochemical methods.

One other concern is that it appears that nanochip expression lowers the levels of CFP in expressing cells (and the representation of CFP expressing cells in the population). This should not be the case and suggests non-specific effects of the nanochips on protein expression.

The nanoCHIP is expressed in a bicistronic P2A vector with CFP such that the fluorescent reporter protein is expressed in a 1:1 ratio with the engineered ligase. Therefore most likely explanation for the observations that CFP is lower for nanoCHIP compared to nano in Figure 2 and Figure 4 is that in these particular transfections we had a comparatively lower expression of nanoCHIP. Please note that this is not the case in Figure 9, for example. It is known (and we have confirmed this) that the vhh4 nanobody recognizes GFP/YFP, but not CFP.

Given these apparent discrepancies greatly strengthen the paper. This would confirm loss of cell surface expression of KCNQ1 by a technique independent of flow cytometry, which is used exclusively to assay levels of cell surface expression (that is, n) independent of effects of nanochips on channel open probability or single channel conductance. Biotinylation analyses could be combined with the type of anti-ubiquitin blots already presented to further support that ubiquitinated proteins are internalized.